

# Molecular composition and processing of aqueous secondary organic aerosol in cloud at a mountain site in southeastern China

Yali Jin[1,2,3,4], Hao Luo[1,6], Siqi Tang[1], Shuhui Xue[1], Chengyu Nie[1], Xiaocong Peng[3], Yan Zheng[9], Weiqi Xu[7], Guohua Zhang[7], Xiaole Pan[8], Yele Sun[8], Qi Chen[9], Lanzhong Liu[8], and Defeng Zhao[1,3,4,5]

[1]Department of Atmospheric and Oceanic Sciences & Institute of Atmospheric Sciences, Fudan University, Shanghai, 200438, China

[2]Shanghai Frontiers Science Center of Atmosphere-Ocean Interaction, Fudan University, Shanghai, 200438, China

[3]Shanghai Key Laboratory of Ocean-land-atmosphere Boundary Dynamics and Climate Change, Fudan University, Shanghai, 200438, China

[4]National Observations and Research Station for Wetland Ecosystems of the Yangtze Estuary, Fudan University, Shanghai, 200438, China

[5]Institute of Eco-Chongming (IEC), 20 Cuiniao Rd., Chongming, Shanghai, 202162, China

[6]Inner Mongolia Key Laboratory of Pollution Control and Low Carbon Resource Utilization, Inner Mongolia University, Inner Mongolia, 010021, China

[7]State Key Laboratory of Advanced Environmental Technology, Guangzhou Institute of Geochemistry, Chinese Academy of Sciences, Guangzhou, 510640, China

[8]State Key Laboratory of Atmospheric Environment and Extreme Meteorology, Institute of Atmospheric Physics, Chinese Academy of Sciences, Beijing, 100029, China

[9]State Key Laboratory of Regional Environment and Sustainability, College of Environmental Sciences and Engineering,
Peking University, Beijing, 100871, China

*Corresponding to*: Defeng Zhao (dfzhao@fudan.edu.cn)



**Abstract.** Aqueous secondary organic aerosol (aqSOA) contributes substantially to organic aerosol (OA), affecting air quality, human health, and climate. However, the molecular composition and processing of aqSOA in cloud remain unclear due to limited online field measurements. We measured molecular composition of OA online (time resolution 20 s) and tracked its

processing at a mountain site in southeastern China, using an Extractive ElectroSpray Ionization inlet coupled with a Time-of-Flight Mass Spectrometer (EESI-ToF-MS). We identified 2084 molecular formulas and compared OA composition from three sample types with adjacent time (<2 h): cloud droplets (CD), interstitial aerosol (INT), and cloud-free aerosol (CF) in representative cloud episodes. CHO class was the dominant constituent, followed by CHON class. The fraction of CHO was lower in CD than that in INT and CF, while the fraction of CHON was higher, which may result from the uptake of

organonitrates or nitration in cloud water. Compounds in CD had more carbon, oxygen, and nitrogen number but lower O/C than INT and CF, which is attributed to accretion reactions in cloud water. We identified aqSOA tracers, including 39 new compounds, which were significantly enriched in CD compared with CF. This study also reveals rapid changes of aqSOA composition, which highlight the necessity for high time resolution measurement to capture the processing of aqSOA in cloud. Overall, this study provides clear information of processing of aqSOA in cloud and highlights the importance of accretion

reactions, which has implications on the composition and physicochemical properties of SOA.



## 1 Introduction

Secondary organic aerosol (SOA) is a major component of organic aerosol (OA) with diverse emission sources, gaseous precursors, and composition, exerting significant impacts on air quality, climate, and human health (Jimenez et al., 2009; Nault

et al., 2021). SOA is primarily produced through the oxidation of volatile organic compounds (VOCs), while the atmospheric aging of primary organic aerosol (POA) may also contribute. Numerous previous studies have investigated the formation mechanisms of SOA, with particular emphasis on gas-phase pathways (Odum et al., 1996; Ervens et al., 2011). However, SOA formed solely through gas-phase reactions (gasSOA) cannot fully account for the observed SOA concentrations (de Gouw et al., 2005; Volkamer et al., 2007; Volkamer et al., 2006). In addition to the traditional gas-phase processing, aqueous-phase

pathways have been recognized as an important source of SOA, as supported by laboratory studies (Ervens et al., 2011; Tan et al., 2009; Zhang et al., 2010) and model simulations (Fu et al., 2008; Lamkaddam et al., 2021).

Mounting evidence for aqueous secondary organic aerosol (aqSOA) has been reported in field observations in various atmospheric aqueous systems, i.e., aerosol liquid water (ALW), fog water, and cloud water. For example, several studies on source apportionment in different sites showed that aqSOA is an important contributor to SOA, with its fraction particularly

elevated (up to 44 %) under high relative humidity (RH) conditions and during foggy or cloudy days (Wang et al., 2021; Zhao et al., 2019; Tong et al., 2021; Gilardoni et al., 2016; Duan et al., 2022; Xu et al., 2019; Sun et al., 2016). In fog water, Duan et al. (2021) identified aqSOA and investigated the contribution and formation processes of aqSOA. Additionally, fog water samples were analyzed and compared with aerosol in OA composition (Brege et al., 2018; Kim et al., 2019; Gilardoni et al., 2016). Fog water and cloud water are both diluted aqueous systems. In contrast to fog, cloud is more common, ubiquitously

presents in the atmosphere, and consists of a large quantity of droplets generated by aerosol activation, providing an aqueous medium for physical processes and chemical reactions (McNeill et al., 2012; McNeill, 2015). Cloud or fog processing affects OA in many aspects, including composition, concentration, size distribution, hygroscopicity, oxidation state, and can form brown carbon such as heterocyclic compounds, thus potentially affect air quality and radiation balance of the atmosphere (Wang et al., 2024; Chen et al., 2024; Jimenez et al., 2009; Altieri et al., 2008; Gramlich et al., 2023; Motos et al., 2019).

Many field campaigns have been conducted to investigate characteristics of aqSOA in cloud droplets. Several previous field campaigns investigated OA formation during cloud processing using aerosol mass spectrometer (AMS) or aerodyne aerosol chemical speciation monitor, which obtained information on fragment ions of compounds, such as the fraction of m/z 44 ($CO_2^+$) in the mass spectra (Dadashazar et al., 2022; Gao et al., 2023; Lance et al., 2020; Karlsson et al., 2022). Other studies applied single-particle mass spectrometry (SPMS) to investigate the composition of aqSOA in cloud droplets (Zhang et

al., 2024a; Lin et al., 2017). However, studies using AMS or SPMS cannot provide molecular information on aqSOA, hindering a detailed understanding of its chemical composition as well as the mechanisms of its formation and transformation.



Although offline analysis using instruments such as Gas Chromatography-Mass Spectrometer (Collett et al., 2008), and Fourier Transform Ion Cyclotron Resonance Mass Spectrometry (FT-ICR-MS) (Brege et al., 2018; Sun et al., 2021; Cook et al., 2017; Pailler et al., 2024; Liu et al., 2023b; Zhao et al., 2013; Bianco et al., 2019) can provide molecular information, the time

resolution of several hours or even one day limited by offline filter sampling is insufficient to capture the variations of aqSOA in cloud. In a word, due to limited time and chemical resolution of previous measurements, it is necessary to obtain online molecular information of aqSOA in cloud to provide insights into the detailed chemical composition and the mechanism of its chemical processes.

To get a detailed understanding of cloud processing of aqSOA, we measured the real-time molecular composition of

aqSOA in cloud using an Extractive ElectroSpray Ionization inlet coupled with a Time-of-Flight Mass Spectrometer (EESI-ToF-MS) in a mountain site in southeastern China. In this study, we identify molecular formulas of OA in cloud processing and compare differences in OA characteristics between cloud droplets (CD), interstitial aerosol particles (INT), and cloud-free aerosol particles (CF). We explore new compounds formed in cloud processing and explain their potential formation mechanisms. We also aim to track the temporal evolution of compounds in aqSOA during cloud processing.

**2 Methods**

We conducted this field campaign from May 1$^{st}$ to May 29$^{th}$ in 2024 at Shanghuang Eco-Environmental Observatory of Chinese Academy of Sciences at the summit of the Damaojian mountain (119.51° E and 28.58° N, 1128 m above sea level) that is located in Jinhua city, Zhejiang province, China. The site is a background monitoring station surrounded by coniferous and broad-leaved forests away from megacities, as shown in Fig. 1. In addition to biogenic emissions, this site may be affected

by anthropogenic activities originating from the surrounding small counties, as mentioned in Zhang et al. (2024b).

Cloud droplets (CD) were collected using a Ground-based Counterflow Virtual Impactor (GCVI, Brechtel Manufacturing Inc., Model 1205). The GCVI collected CD with diameters larger than 8.5 μm (Shingler et al., 2012) under conditions of visibility < 3 km, RH > 95 %, and absence of precipitation. After separation from INT (non-activated aerosol in cloud), the CD were dried by mild heating (40 °C) within the GCVI (Lin et al., 2017) and further by a Nafion dryer downstream, and the

residues of CD were subsequently measured. We note that the term "CD" in the Results and Discussion section refers to the residues of cloud droplets. Because the focus is on the relative compositional change of OA in CD and CF, the GCVI enhancement factor was not applied. A PM$_{2.5}$ (particulate matter smaller than 2.5 μm) cyclone inlet (URG, USA) was used to collect INT and CF. A switching system alternated between the GCVI and the URG inlet: PM$_{2.5}$ was sampled when GCVI detected no cloud, whereas CD sampling was triggered automatically once cloud presence was detected by GCVI. During



cloud episodes, the switch was also configured to alternate between CD and INT sampling. It should be noted that the terms

"cloudy days" and "cloudless days" in this study specifically refer to periods with and without low clouds.

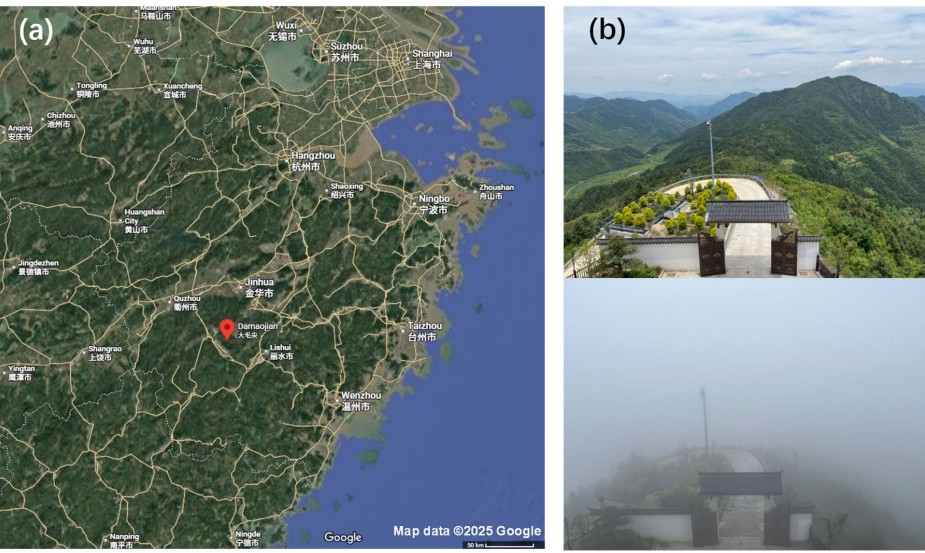

**Figure 1. Location of the Shanghuang site. (a) a map (from © Google Maps) and (b) two photos of sampling site, one with cloud and another without cloud.**

Measurements of CD, INT, and CF were conducted through a manifold positioned downstream of both the GCVI and

URG inlets. The concentrations of OA composition were measured online using EESI-ToF-MS (Aerodyne Institute) with a

time resolution of 20 s. This mass spectrometer achieves soft ionization while preserving the structure of compounds,

measuring molecular formulas with high mass resolution (8000–10000) and low detection limit. Detailed information

regarding EESI-ToF-MS has been reported previously in Lopez-Hilfiker et al. (2019) and our previous studies (Luo et al.,

2024) and (Xue et al., 2025). Here is a brief introduction. Aerosol was sampled after gaseous compounds were removed by

entering a charcoal denuder, and subsequently intersected with an electrospray generated from a working solution containing

100 ppm NaI in a 1:1 (v/v) water and acetonitrile mixture, allowing aerosol compounds to be detected as [M+Na$^+$] in positive

ion mode. Background measurements were obtained by switching the inlet to a filter. The durations of sample and background

collection can be adjusted to ensure aerosol signal levels return to baseline within the time of background (Qi et al., 2019). In

this campaign, sample and background were set in combinations of 10 min and 5 min typically. The sampling volume of EESI-

ToF-MS was 0.9 L m$^{-3}$. Weekly calibration was performed using levoglucosan, and the sensitivity was assumed identical for

all compounds. This assumption does not affect our results since we specifically focus on relative compositional changes of

OA in CD and CF as mentioned above. All organic compound signals are shown as relative intensities normalized to (NaI)Na$^+$

to avoid interference from the ion source fluctuations in EESI-ToF-MS. Mass spectral data were processed using Tofware

3.2.5 in Igor Pro 8. For data screening, the signal to background ratio (s/b) was calculated as the median value of (sample




signal−background)/background, thereby excluding compounds showing insignificant differences between sample and background. Only compounds with the s/b ratio greater than 0.1 were included (Tong et al., 2021). After this screening, 79, 148, 604, and 126 compounds were retained from cloud episodes one to four, respectively. All results presented in Sect. 3.2 are based on these screened data.

The OA size distribution was characterized using a scanning mobility particle sizer (SMPS, TSI 3936), which, together with an atomizer (TSI, 3076), was used to facilitate EESI-ToF-MS calibration. $PM_{2.5}$ concentration was monitored using a Thermo Scientific instrument (Thermo Scientific. Model 5014i), while CO was measured by a Picarro greenhouse gas analyzer (Picarro Inc., G2401). Meteorological parameters including RH, Temperature (T), wind speed (WS), and wind direction (WD) were monitored by an automatic weather station.

The 72 h backward trajectories of air masses arriving at the Shanghuang site were calculated by Hybrid Single-Particle Lagrangian Integrated Trajectory (HYSPLIT) model, with Global Data Assimilation System (GDAS) meteorological data at 1°×1° spatial resolution (Stein et al., 2015; Rolph et al., 2017). These trajectories were clustered into several appropriate groups for 4 CEs. The clustering was based on the total spatial variance (TSV) method (Song et al., 2023; Roland et al., 2025).

## 3 Results and discussion

### 3.1 Characteristics of cloud episodes

During the entire campaign, $PM_{2.5}$ concentration was 13.4±10.8 μg m$^{-3}$ (mean value ± standard deviation), with the highest concentration of 72.2 μg m$^{-3}$ observed on cloudless days, as shown in Fig. 2. The $PM_{2.5}$ concentration is typical in rural areas of China and is substantially lower than that observed in the same season in metropolitan cities such as Beijing (~40 μg m$^{-3}$) and Shanghai (~30 μg m$^{-3}$) (Liu et al., 2023a; Yin et al., 2023).

Cloud episodes accounted for 27.1 % of the one-month campaign, with the sample types of CD and INT representing 13.1 % and 14.0 %, respectively. Of the 16 recorded cloud episodes, we selected those without precipitation and with adjacent cloud-free periods (<2 h from CD) to avoid the influence of wet deposition and ensure CD/INT/CF sampling coverage. Six out of 16 episodes meet these criteria, and four cloud episodes (CEs) are further selected. CE1, CE2, CE3, and CE4 differ in $PM_{2.5}$ and CO concentration, meteorological conditions, origin of air mass, and duration time, as shown in Table S1. The duration of each CE ranged from several minutes to three days. Meteorological conditions and origin of air masses are discussed in Text S1, and backward trajectories from HYSPLIT are shown in Fig. S2.



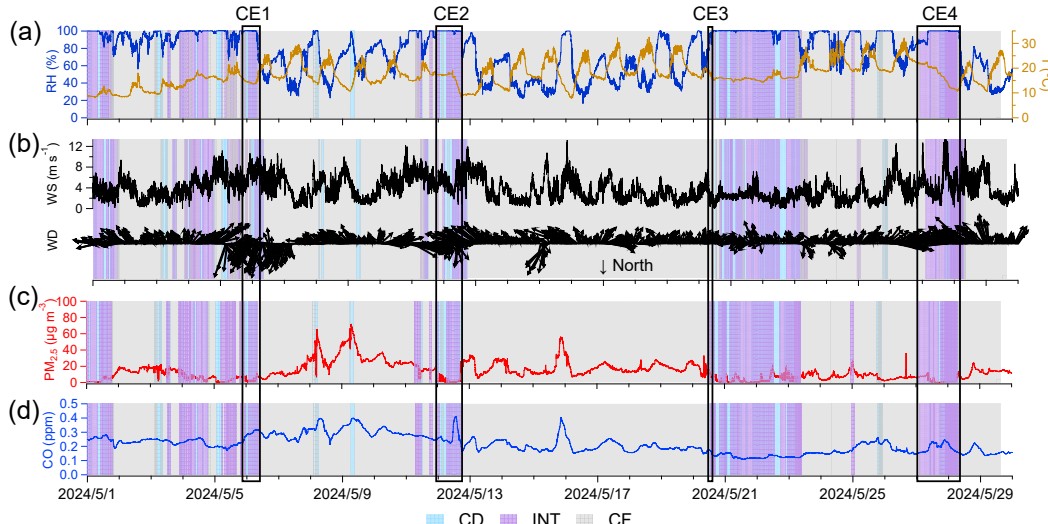

**Figure 2. Time series of (a) RH and T, (b) WS and WD, (c) PM$_{2.5}$, and (d) CO. Sample types of cloud droplets (CD), interstitial**

**aerosol particles (INT), and cloud-free aerosol particles (CF) are shaded as blue, purple, and gray, respectively.**

Each CE's sampling period is divided into three stages (pre-cloud, in-cloud, and post-cloud) to compare OA characteristics. The in-cloud stage corresponds to the sample types of CD and INT, whereas both the pre-cloud and post-cloud stages correspond to the sample type of CF (PM$_{2.5}$). Detailed characteristics of sample types in four CEs, such as mean chemical formula, H/C, O/C, N/C, and OSc (carbon oxidation states, 2×O/C−H/C), are shown in Table 1, and division of stages is shown

in Fig. S1.

A total of 2084 molecular formulas of OA were identified in the campaign. Mean formula of CD was C$_{10.01–12.81}$H$_{14.59–}$ $_{20.34}$O$_{5.08–6.00}$N$_{0.34–0.43}$S$_{0–0.01}$Si$_{0–1.07}$ for CE1–CE4. Compared with pre-cloud aerosols with formula C$_{8.43–11.10}$H$_{14.23–16.83}$O$_{5.06–}$ $_{5.72}$N$_{0.16–0.35}$S$_{0–0.01}$Si$_{0–0.28}$, CD exhibited increased numbers of carbon, hydrogen, oxygen, and nitrogen atoms. These molecular formulas were classified into eight classes, that is, CHO (only C, H, O atoms are contained in the chemical formula, hereafter),

CHON, CHONS, CHOS, CHN, CHS, CHNS, and CHOSi. Since the composition of OA varied in different CEs, the fractions of these OA classes are discussed for each CE in Sect. 3.2. The O/C ratio was generally lower in CD (0.45–0.66) than in pre-cloud aerosols, INT, and post-cloud aerosols in 4 CEs. The O/C ratio in CD is comparable to those reported for fog water (0.52–0.68), aqSOA (0.61–0.84), and oxygenated OA (0.44–0.83) by Gilardoni et al. (2016). In general, O/C of CD in this study is comparable to that of fog (0.58–0.8) in the Po Valley in Brege et al. (2018), while H/C of CD (1.49–1.82) is lower

than that of fog (1.29–1.37) in that study. Furthermore, CD showed elevated N/C (0.034–0.047) relative to other sample types, while its OSc value (−0.72 to −0.24) is generally lower than in other sample types.





**Table 1. Detailed characteristics of OA chemical composition for CE1–CE4 in pre-cloud aerosols, CD, INT, and post-cloud aerosols. Shown are the mean chemical formulas and the mean H/C, O/C, N/C, and OSc values.**

| | | Pre-cloud aerosols | CD | INT | Post-cloud aerosols |
|---|---|---|---|---|---|
| CE1 | Formula | $C_{9.10}H_{14.37}O_{5.72}N_{0.24}S_0Si_{0.28}$ | $C_{10.45}H_{16.69}O_{6.00}N_{0.34}S_0Si_{0.55}$ | $C_{9.20}H_{14.41}O_{5.72}N_{0.27}S_0Si_{0.24}$ | $C_{9.28}H_{14.73}O_{5.79}N_{0.26}S_0Si_{0.36}$ |
| | H/C | 1.60 | 1.60 | 1.59 | 1.59 |
| | O/C | 0.70 | 0.66 | 0.69 | 0.69 |
| | N/C | 0.025 | 0.034 | 0.028 | 0.028 |
| | OSc | −0.21 | −0.29 | −0.21 | −0.20 |
| CE2 | Formula | $C_{8.62}H_{14.23}O_{5.06}N_{0.31}S_{0.01}Si_{0.21}$ | $C_{10.16}H_{18.84}O_{5.45}N_{0.41}S_{0.01}Si_{1.07}$ | $C_{8.79}H_{14.56}O_{5.00}N_{0.35}S_{0.01}Si_{0.22}$ | $C_{11.04}H_{19.73}O_{6.14}N_{0.38}S_{0.01}Si_{1.14}$ |
| | H/C | 1.69 | 1.82 | 1.70 | 1.71 |
| | O/C | 0.63 | 0.58 | 0.61 | 0.60 |
| | N/C | 0.038 | 0.047 | 0.043 | 0.037 |
| | OSc | −0.44 | −0.66 | −0.49 | −0.50 |
| CE3 | Formula | $C_{11.10}H_{16.83}O_{5.07}N_{0.35}S_{0.01}Si_{0.03}$ | $C_{12.81}H_{20.34}O_{5.08}N_{0.43}S_{0.01}Si_{0.46}$ | $C_{10.88}H_{16.77}O_{5.07}N_{0.31}S_{0.005}Si_{0.10}$ | $C_{11.06}H_{16.84}O_{5.09}N_{0.32}S_{0.01}Si_{0.06}$ |
| | H/C | 1.56 | 1.63 | 1.58 | 1.56 |
| | O/C | 0.53 | 0.45 | 0.53 | 0.53 |
| | N/C | 0.036 | 0.041 | 0.031 | 0.033 |
| | OSc | −0.51 | −0.72 | −0.52 | −0.50 |
| CE4 | Formula | $C_{8.43}H_{13.27}O_{5.36}N_{0.16}S_{0.004}Si_0$ | $C_{10.01}H_{14.59}O_{5.72}N_{0.34}S_{0.01}Si_0$ | $C_{9.13}H_{13.56}O_{5.56}N_{0.23}S_{0.003}Si_0$ | $C_{8.94}H_{13.36}O_{5.57}N_{0.20}S_{0.003}Si_0$ |
| | H/C | 1.64 | 1.49 | 1.51 | 1.52 |
| | O/C | 0.68 | 0.62 | 0.65 | 0.67 |
| | N/C | 0.018 | 0.034 | 0.024 | 0.022 |
| | OSc | −0.28 | −0.24 | −0.20 | −0.18 |


## 3.2 Comparison of cloud episodes

The fractions of three OA classes which are CHO, CHON, and Others (including CHONS, CHOS, CHN, CHS, CHNS, and CHOSi) exhibited general similarities across the four CEs, as shown in Fig. 3. In all sample types (pre-cloud aerosols, CD, INT, and post-cloud aerosols) of the four CEs, CHO dominated OA composition, accounting for > 50 % of OA (54.6 %–85.7 %

from CE1 to CE4), followed by CHON (14.0 %–33.2 %) and Others (lower than 18.7 %). The Others class, predominantly CHOSi, accounted for 0.5 %–18.7 % in CD, exceeding the fractions in other sample types, and is further discussed at the level of individual compounds below. Generally, CD showed the lowest CHO fraction (54.8 %–70.7 %) and the highest CHON fraction (26.6 %–33.2 %) among the four sample types. The only exception in CE2, with higher CHON (29.4 %) and lower CHO (54.6 %) in post-cloud aerosols than other sample types, is attributable to air mass changes during the long time interval

between post-cloud and others (shown in Fig. 2 and Fig. S1), as indicated by elevated CO concentration. CHON compounds have been detected in cloud and fog water in numerous studies (LeClair et al., 2012; Sun et al., 2024a; Sun et al., 2021; Sun et al., 2024b). Higher CHON fraction (59.0 %–63.5 %) in fog water than aerosol particles (51.2 %–51.5 %) was reported previously in Sun et al. (2024a), which is in agreement with the results in this study. In addition, the greater number of CHON compounds in CD compared with CF underscores the role of cloud processing in enhancing CHON, as reflected in number

fraction rather than intensity fraction (Boone et al., 2015; Liu et al., 2023b). The fraction of CHON (19.7 %–26.3 %) in INT



was lower than in CD (26.6 %–33.2 %) and higher than in pre-cloud aerosols (14.0 %–23.1 %) in CE1, CE2, and CE4. However, in CE3, the slightly lower CHON fraction in INT compared to pre-cloud aerosols may be due to fluctuation in cloud resulting from the short duration time of the cloud (several minutes). Higher relative abundance of CHON in CD (43.6 %–65.3 %) compared to INT (31.8 %–51.0 %) has been observed at Tianjing Mountain in southern China (Sun et al., 2021), consistent

with our results. The higher CHON fraction in CD than in pre-cloud aerosols suggests that cloud processing promoted CHON formation. Higher CHON in INT compared to pre-cloud aerosols indicates that although not activated into cloud droplets, high RH experienced by INT (close to 100 %) and corresponding high aerosol water content could still promote CHON formation in INT, consistent with the elevated N/C ratio of aqSOA of aerosol particles under high RH conditions (Zhao et al., 2019).

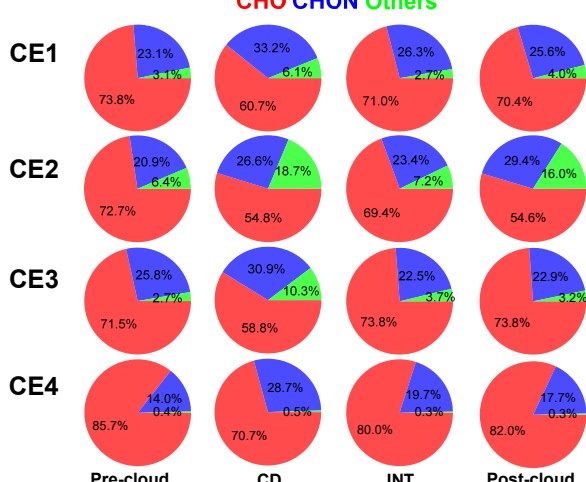

**Figure 3. Fractions of three classes of OA, namely CHO, CHON, and Others (including CHONS, CHOS, CHN, CHS, CHNS, and CHOSi) in pre-cloud aerosols, CD, INT, and post-cloud aerosols in CE1, CE2, CE3, and CE4.**

Among the CHON class, the compounds enriched in CD, such as $C_{8–12}H_{11–19}NO_{5–8}$, $C_{14–16}H_{21–27}NO_{4–9}$, with an O/N ratio of ≥3 (69 %–88 % for CE1–4), suggesting that they are likely organonitrates, amino acids, or nitrogen-heterocyclic compounds. At the Shanghuang site, emissions of monoterpenes and sesquiterpenes are abundant (Zhang et al., 2024b). Consequently,

$C_{10}H_{15}NO_x$ and $C_{10}H_{17}NO_x$ may be formed via hydroxyl oxidation of monoterpene in the presence of NO (Shen et al., 2022) or $NO_3$ oxidation (Shen et al., 2021; Guo et al., 2022) and dissolve in the aqueous phase, whereas $C_{15}H_{23}NO_x$ and $C_{15}H_{25}NO_x$ may originate from similar reactions involving sesquiterpenes. Additionally, precursors could form organonitrates through aqueous reactions, e.g., with $NO_3$ radicals (Ng et al., 2017), or involving $NO_3^-$ (Sun et al., 2024b; Huang et al., 2023; Barber et al., 2024). These reactions can occur at night or even during the day under reduced light conditions in cloud. This finding

contrasts with the observation at Mt. Tai, where, despite the higher number of CHON compounds in CD relative to CF, a larger fraction contained reduced nitrogen groups (O/N <3) (Liu et al., 2023b). Such disparity may arise from differences in



precursors between the two sampling sites. Additional information, such as the gas-phase CHON composition and concentration, is required to further elucidate the formation mechanisms of these compounds.

The molecular composition characteristics of OA in four CEs exhibit similar patterns, presented as carbon number distribution colored according to the numbers of oxygen and nitrogen; therefore, only CE2 and CE3 are shown in Fig. 4 (CE1 and CE4 in Fig. S3). In each CE, comparison is carried out between CD and CF which was the closest to CD temporally: for CE1–3, CD is compared with pre-cloud aerosols, while CE4 is compared with post-cloud aerosols, as shown in Fig. S1. In CD, the carbon number of OA ranged from 2 to 28 (CE2: 3–27; CE3: 2–28), and the oxygen number ranged from 0 to 10 in CE2 and 0 to 15 in CE3. Comparing with OA in CF, OA in CD contained higher fraction of compounds with $n_C > 10$ as well as elevated $n_O$ (CE2: $n_O$=7–10; CE3: $n_O$=6–15).

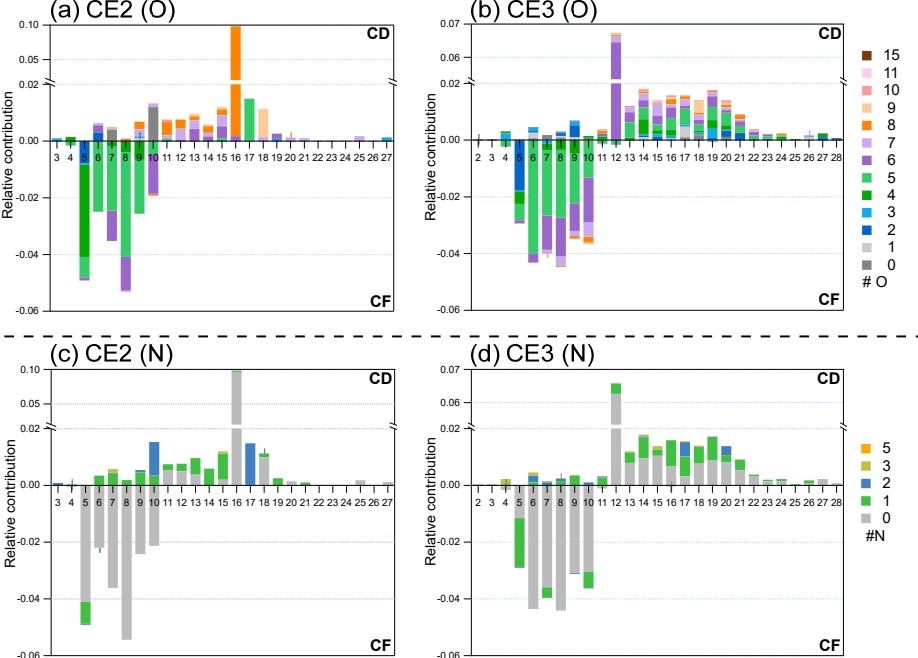

**Figure 4. Detailed relative contribution of OA. The average carbon number distribution of differences between CD and CF are colored by oxygen number of (a) CE2, (b) CE3; and nitrogen number of (c) CE2, (d) CE3. Positive value stands for significant molecular characteristics of CD, and negative value stands for that of CF. Fractions of compounds are normalized to sum of signals of all organics in CD and CF, respectively.**

The nitrogen number ($n_N$) distributions relative to $n_C$ exhibit similar patterns in all CEs. In CE2, the $n_N$ of N-containing OA is distributed from 0 to 3, and from 0 to 5 in CE3. The $n_C$ of N-containing OA ranged from 3 to 21 in CE2 and 4 to 27 in CE3. Compared with CF, CD contained a larger fraction of N-containing OA, especially those with $n_N$=1–3 and higher $n_C$. Collectively, compounds in CD had more $n_C$, $n_O$, and $n_N$ than those in CF. These molecular characteristics are likely attributed





to accretion reactions such as oligomerization (Yu et al., 2016). This finding is consistent with several laboratory studies of

aqSOA formation. For instance, enriched high molecular weight compounds in aqSOA were reported in the bulk phase

experiments of methylglyoxal and glyoxal under cloud-relevant conditions (Tan et al., 2009; Altieri et al., 2008). And aqSOA

from in-cloud simulation using a wetted-wall flow reactor has more highly oxygenated and carbon-containing compounds than

gasSOA simulated by an oxidation flow reactor (OFR) from the same biomass burning samples (Wang et al., 2024).

Experiments in the bulk phase and the wetted-wall flow reactor which better represents atmospheric aqueous conditions,

indicate that accretion reactions could be prevalent in cloud droplets.

     To investigate OA processing when the cloud episode changed from CF to CD, $CH_2$-based Kendrick mass defect (KMD)

plots for CE2 and CE3 are analyzed (Fig. 5). The chemical formulas of compounds with a larger fraction in CD than CF in

four CEs are listed in Table S2. Several series of compounds in CE2 and CE3 exhibit sequential increase in $CH_2$ groups, such

as $C_{10}H_{15}NO_6(CH_2)_n$, $C_8H_{11}NO_7(CH_2)_n$, $C_4H_6O_5(CH_2)_n$, $C_5H_6O_5(CH_2)_n$, $C_5H_6O_6(CH_2)_n$, $C_7H_8O_6(CH_2)_n$. Specifically, numerous

CHON compounds were present at higher fractions in CD, with some labeled by formulas such as series of $C_6H_9NO_7(CH_2)_{0-4}$,

$C_7H_{11}NO_6(CH_2)_{0,2,3}$, $C_8H_{11}NO_7(CH_2)_{0-5,7}$, and $C_{12}H_{17}NO_8(CH_2)_{0-3}$ in CE2 and series of $C_7H_9NO_4(CH_2)_{0,3,7,9}$,

$C_8H_{11}NO_7(CH_2)_{0-3,6,8-9,12}$, $C_{10}H_{15}NO_6(CH_2)_{0-4,6}$, $C_{12}H_{19}NO_5(CH_2)_{0,2-3}$ in CE3. This result is in agreement with the higher

fraction of total CHON compounds in CD compared with CF, as discussed above. For most homologues, CD contained higher

fractions of larger compounds (with more $CH_2$ groups) than CF, while lower fractions of smaller compounds. As detailed

above, it is likely that cloud processing enhanced accretion reactions by extending the length of the carbon chain, which further

highlights the importance of accretion reactions of organics in cloud droplets. In CF, CHO had a larger fraction than CHON;

for example, CHO compounds such as $C_5H_6O_6(CH_2)_n$, $C_6H_{10}O_6(CH_2)_n$, and $C_5H_6O_5(CH_2)_n$ were more abundant. The pattern

of adding $CH_2$ groups in cloud processing is similar in all CEs. However, the KMD plot based on O shows that compounds in

CE2 and CE3 did not exhibit a clear pattern with a sequential increase in O (Fig. S4). The dominant pattern of $CH_2$ addition,

rather than O addition, suggests that sequential OH addition or auto-oxidation was not prevalent in cloud processing. In terms

of the increments of $CH_2$ and O, $CH_2$ displays a wider growth trend (0–7) among all series, whereas O shows a narrower

increase, confined to a range of 0 to 3. Consequently, results of KMD plots suggest that as cloud processing proceeded, $n_C$ of

OA increases, while the increase in $n_O$ is lower than the $n_C$, agreeing with the lower O/C ratio in CD than that in CF. The

possible reason is that aqueous processing is more significant in accretion (enhancing $n_C$) than oxygenation (enhancing $n_O$).

     Some siloxane compounds showed higher fractions in CD than in CF, such as $C_{16}H_{48}O_8Si_8$ (m/z 615) and $C_{18}H_{54}O_9Si_9$

(m/z 689) in CE2 and $C_{12}H_{36}O_6Si_6$ (m/z 467) and $C_{18}H_{54}O_9Si_9$ in CE3. Siloxane is a type of volatile chemical products such as

those found in personal care products (Gkatzelis et al., 2021; McDonald et al., 2018). To our best knowledge, this is the first

time that $C_{16}H_{48}O_8Si_8$ was observed in cloud droplets. The reason for the higher fraction of siloxane warrants further study.






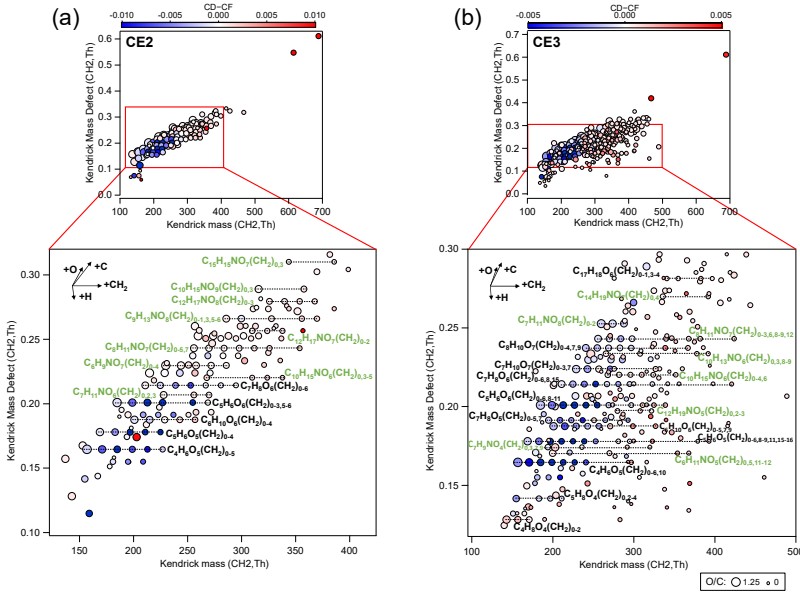

**Figure 5. Kendrick mass defect plot based on CH₂ of compounds in (a) CE2 and (b) CE3. Data points are color-coded by differences of fractions of compounds between CD and CF and sized by the O/C ratio. Fractions of compounds are normalized to the sum of signals of all organics in CD and CF, respectively. Note that, for conciseness, data points in CE3 with normalized signal difference**

**between −0.0003 and 0.0003 (appearing nearly white) are not shown here. The molecular formulas include the reagent ion Na⁺, which is not shown for simplicity.**

### 3.3 Characteristic compounds in cloud processing and formation mechanisms

We identified aqSOA tracers in cloud droplets by comparing the intensity fractions of all compounds between CD and CF using a *t*-test at a significant level of 0.05. A total of 144, 421, 274, and 537 organic compounds in CE1, CE2, CE3, and

CE4, respectively, passed the *t*-test. Among these compounds, 39 organic compounds in CD were significantly enriched in three or four CEs, as shown in Table 2. Two were consistently significant in CD across all four CEs: $C_{14}H_{42}O_7Si_7$ and $C_9H_{22}N_2O_4$. Furthermore, sulfate compounds were enriched in CD compared with CF in three CEs. Sulfate is a well-established tracer for aqueous-phase processing, and its elevated concentration in cloud droplets and fog has been widely reported (Dadashazar et al., 2022; Brege et al., 2018; Kim et al., 2019). This result further confirms the reliability of the identification

of aqSOA tracers. Accordingly, these 39 OA compounds can be regarded as aqSOA tracers. The number of CHO, CHON, CHN, and CHOSi compounds is 15, 19, 2, and 3, respectively. The majority of tracers exhibit carbon numbers greater than nine, which is also an indication of accretion reactions in cloud droplets. Most of these aqSOA tracers have not been reported in previous literature (Cook et al., 2017; Bianco et al., 2019).




**Table 2. Thirty-nine aqSOA tracers observed in CD of three or four CEs. These tracers are classified into four classes: CHO, CHON, CHN, and CHOSi.**

| CHO | CHON | CHN | CHOSi |
|---|---|---|---|
| $C_6H_{12}O_6$ | $C_4H_7NO_4$ | $C_9H_{18}N_2$ | $C_{12}H_{36}O_6Si_6$ |
| $C_{10}H_{16}O_2$ | $C_5H_{11}N_2O$ | $C_{12}H_{23}N$ | $C_{14}H_{42}O_7Si_7$ |
| $C_{10}H_{22}O_4$ | $C_9H_{22}N_2O_4$ | | $C_{16}H_{48}O_8Si_8$ |
| $C_{12}H_{26}O_5$ | $C_9H_{13}NO_2$ | | |
| $C_{13}H_{22}O$ | $C_{10}H_{19}NO$ | | |
| $C_{13}H_{26}O_5$ | $C_{10}H_{19}NO_3$ | | |
| $C_{15}H_{24}O_{14}$ | $C_{11}H_{19}NO_4$ | | |
| $C_{15}H_{26}O_8$ | $C_{13}H_{30}N_2O_4$ | | |
| $C_{15}H_{32}O_6$ | $C_{13}H_{23}NO_3$ | | |
| $C_{16}H_{30}O_4$ | $C_{14}H_{29}N_3O_3$ | | |
| $C_{20}H_{22}O_5$ | $C_{14}H_{29}NO_4$ | | |
| $C_{21}H_{36}O_8$ | $C_{18}H_{33}NO_5$ | | |
| $C_{23}H_{44}O_3$ | $C_{18}H_{29}NO_5$ | | |
| $C_{24}H_{40}O_3$ | $C_{19}H_{37}NO_3$ | | |
| $C_{30}H_{56}O_2$ | $C_{20}H_{29}NO_5$ | | |
| | $C_{21}H_{41}NO_2$ | | |
| | $C_{22}H_{34}N_2O_6$ | | |
| | $C_{29}H_{51}NO_2$ | | |
| | $C_{29}H_{51}NO_6$ | | |

Furthermore, 236 OA compounds were significantly enriched in two of four CEs, including the common aqSOA tracer, oxalic acid ($C_2H_2O_4$) previously reported in field observations and laboratory studies (Rogers et al., 2025; Ervens et al., 2011).

The compound $C_2O_4Na_3^+$ is identified as oxalic acid, of which the hydrogen atoms in the carboxylic functional group (-COOH) are substituted by $Na^+$ (Surdu et al., 2024). The oxalic acid signal was exclusively observed during CD, whereas it remained weak and noisy in CF and INT, as shown in Fig. 6. Overall, the oxalic acid signal was not significant in all four CEs, primarily attributed to large fluctuations. Meanwhile, the $C_6H_{12}O_6$ signal was as low as the detection limit in CF; however, it increased gradually when CD began. $C_6H_{12}O_6$ in aqueous formation was reported in a laboratory study and may be produced from the

aqueous reaction of formaldehyde or acetaldehyde (Li et al., 2011). Therefore, it is reasonable to classify $C_6H_{12}O_6$ as a tracer of aqSOA.

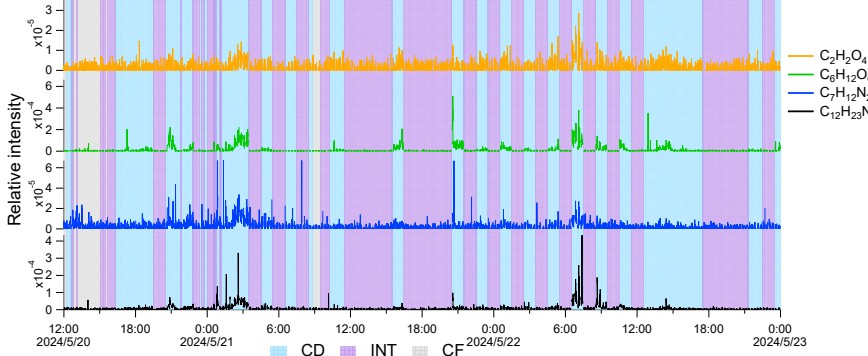

**Figure 6. Time series of compounds: carboxylic acid ($C_2H_2O_4$), $C_6H_{12}O_6$, nitrogen-containing substances including $C_7H_{12}N_2$ and $C_{12}H_{23}N$, in certain periods. CD, INT, and CF are shaded as blue, purple, and gray, respectively. Blank gaps denote background**

**which is not shown.**

Notably, N-containing compounds were significant in CD, such as $C_{12}H_{23}N$ and $C_7H_{12}N_2$. $C_{12}H_{23}N$ may be a compound with a pyrrole structure. Pyrrole-derived SOA may contribute to brown carbon chromophore and influence radiative forcing (Chen et al., 2024). $C_7H_{12}N_2$ (enriched in CE2) is likely 1-butylimidazole, a derivative of imidazole, reported in reactions of methylglyoxal and amines in cloud simulation in De Haan et al. (2011). This compound has been observed from emissions of

residential cooking and agricultural residual burning (Fleming et al., 2018; Wang et al., 2017; Lin et al., 2012). Moreover, imidazole has been reported as a type of brown carbon influencing regional radiative forcing (Kim et al., 2019; Lian et al., 2020) and may contribute to reactive oxygenated species, potentially relating to adverse health effects (Dou et al., 2015). The enhanced concentration of N-containing compounds in cloud droplets could therefore have significant atmospheric implications and warrants further investigation.

**3.4 Dynamic variation of OA in cloud**

Relatively stable T, wind speed, and CO concentration in a typical 3-day cloud indicate this cloud was stable and sources of primary emission remained relatively constant during the whole cloud episode, as shown in Fig. 7. CHO and CHON were the major constituents for most time of the episode, whereas Others was the lowest. The O/N ratio was generally lower in CD than in INT, while he ratio of O/C and N/C varied irregularly in CD and INT. Although the time resolution of our measurement

(~20 s) is enough to capture the evolution of a compound in cloud, either in CD or INT, there was no clear trend in the time series of the compounds, either from the fractions of OA classes or elemental ratios during the sample types of CD or INT. This phenomenon is likely due to the dynamic characteristics of cloud, in which turbulence and chemical processes continuously induced rapid changes in organic compounds, resulting in no gradual trends in their concentrations.

From the perspective of the molecular composition, the relative intensities of representative compounds in cloud episodes

exhibited frequent and pronounced fluctuations during individual CD periods, as shown in Fig. 6. Even within a 1-h CD, the signal of compounds increased and decreased irregularly, likely due to turbulence. Consequently, it is difficult to track and capture information on the chemical transformation of OA in cloud. Most previous comparisons of the chemical composition of cloud droplets with cloud-free aerosol particles or interstitial aerosol particles are based on long sampling (hours to day) and offline analysis (Brege et al., 2018; Sun et al., 2021). Based on the findings in this study, the results obtained using methods

with low time resolution may be subject to uncertainties due to the dynamic nature of clouds.



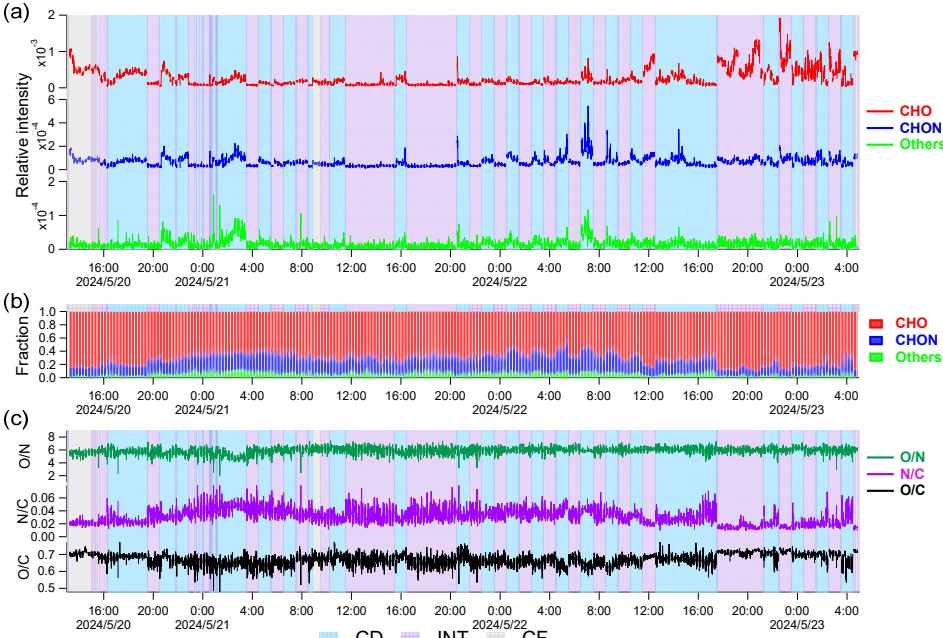

**Figure 7. Time series of three classes of organic compounds (CHO, CHON and Others) in a long cloud. (a) relative intensity, (b) fraction in OA, (c) O/N, N/C, and O/C ratio of OA. Blank gaps denote background data of EESI-ToF-MS, which is not shown. CD, INT, and CF are shaded in blue, purple, and gray, respectively.**

## 4 Conclusions and implications

In this study, we investigated aqSOA molecular composition and processing in cloud episodes using online molecular information (time resolution of 20 s) obtained by EESI-ToF-MS at a high-mountain site in China. Among various classes of compounds, CHO compounds contributed predominantly to OA in cloud droplets. CHON was enhanced markedly in cloud droplets compared with cloud-free aerosol particles and interstitial aerosol particles. The majority of CHON compounds were likely organonitrates, highlighting the enrichment of these compounds in cloud processing. Organics in cloud droplets had an average molecular formula $C_{10.01-12.81}H_{14.59-20.34}O_{5.08-6.00}N_{0.34-0.43}S_{0-0.01}Si_{0-1.07}$ for the selected four cloud episodes. OA in CD had more carbon, oxygen, and nitrogen numbers compared to adjacent cloud-free aerosol particles. Organics in cloud droplets showed a homologue pattern with increasing $CH_2$, and larger compounds (with higher carbon number) were enriched in cloud droplets compared with cloud-free aerosol particles, indicating the importance of accretion reactions in cloud processing of OA.

We identified several compounds that were significantly enriched in cloud droplets, which include some typical aqSOA tracers such as oxalic acid and new compounds such as $C_6H_{12}O_6$, $C_9H_{22}N_2O_4$, and $C_{12}H_{23}N$, etc., that can be used as aqSOA tracers. Nitrogen-containing compounds, including $C_7H_{12}N_2$ and $C_{12}H_{23}N$, were observed to be enriched in cloud droplets



compared with cloud-free aerosol particles. Besides, cloud processing substantially influences OA composition, resulting in large difference among distinct CEs. Based on measurement of high time resolution (~20 s), we find that the concentrations of individual organic compounds were highly dynamic in cloud, which is likely due to the turbulence in cloud. Such a highly dynamic nature in cloud poses difficulties in extracting the influence of chemical processes on individual compounds for instrumentation with low time resolution.

Our study highlights the importance of accretion reactions in cloud processing of OA. Due to the increase in the large molecular weight compounds, accretion reactions likely reduce the volatility of organics and could potentially enhance OA mass concentration and alter the aerosol size distribution after cloud evaporation. The formation of larger compounds can also modify other physicochemical properties, such as lifetime, oxidation state, viscosity, and hygroscopic properties, which may further influence the cloud activation of these aerosols. In addition, the formation of N-containing compounds in cloud droplets,

such as organonitrates, pyrrole, and imidazole, may also affect the physicochemical properties of aqSOA, e.g., contributing to brown carbon and thus affecting regional radiative forcing.

The new tracers of cloud processing found in this study, such as $C_6H_{12}O_6$, could help future studies identify aqSOA processed from cloud processing. Moreover, our results highlight the necessity of high time resolution measurements (< 1 h), especially online measurements (time resolution of minutes) of cloud droplets to investigate the chemical processes in cloud,

considering dynamic variations of compounds in cloud due to turbulence and changes in air masses.



**Data availability.** The data used in this study are available from the corresponding authors upon request: Defeng Zhao (dfzhao@fudan.edu.cn).

**Supplement.**

**Author contributions.** DZ conceptualized the research. YJ conducted the measurements with the aid of HL, DZ, ST, SX, and
CN. XiaocP and GZ conducted GCVI measurements. XiaolP conducted the meteorological measurements. WX, YZ, YS, QC and LL provided support for sampling and operation of the Shanghuang site. YJ processed data and wrote the manuscript. YJ and DZ edited the manuscript with the inputs of all authors.

**Competing interests.** Qi Chen is a member of the editorial board of ACP.

**Financial support.** This work is supported by the National Natural Science Foundation of China (No. 42575109), Shanghai
Pilot Program for Basic Research-Fudan University (No. 21TQ1400100 (22TQ010)), and the National Natural Science Foundation of China (No. 42330605).

**Acknowledgement**. The authors gratefully acknowledge the NOAA Air Resources Laboratory (ARL) for the provision of the HYSPLIT transport and dispersion model and/or READY website (https://www.ready.noaa.gov) used in this publication.

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
