# Peer review of "Molecular composition and processing of aqueous secondary organic aerosol in cloud at a mountain site in southeastern China"

_EGUsphere, 2025_

## Author Comment (AC1)

**Responses to all reviewers:**

We thank the reviewers for the careful review of our manuscript. The comments and suggestions are greatly appreciated and have helped to further improve the manuscript. As shown below, all the comments have been addressed. In the following, please find our responses to the comments and the corresponding revisions made to the manuscript. The point-by-point comments are addressed in the following text. The original comments are shown in italics and responses are in normal font. The revised parts of the manuscript are in blue. The line numbers here refer to the clean version.

**Reviewer: 1**

*This manuscript reports high–time/mass resolution online molecular measurements of OA using EESI-TOF at a mountain site in SE China, comparing cloud droplets, interstitial, and cloud-free aerosol. It is clearly within ACP's scope, addressing atmospheric composition and processes with implications for aerosol–cloud interactions and climate. However, the discussion of "tracers" requires more caution: the manuscript should clearly distinguish between true tracers and compounds showing enhanced signals, define the concept explicitly as illustrated below.*

*Major comments:*

*1 . Introduction: the current motivation for a cloud-droplet study is underdeveloped. The Introduction should more clearly articulate (i) why in-cloud processing matters for OA burdens and properties, (ii) what key uncertainties remain despite prior fog/cloud and online/offline molecular studies, and (iii) why an online, molecular-level approach (EESI-TOF) is uniquely positioned to address those gaps. Please sharpen the problem statement, specify the hypotheses, and state the concrete research questions and expected contributions of this dataset.*

**Response:** Accepted.

(i)As suggested, in the revise manuscript we have further sharpen the problem statement and added more illustrations on the motivation of our cloud-droplet study in line 56-63 in the manuscript as follows:

"In contrast to fog, cloud is more common, ubiquitously presents in the atmosphere, and consists of a large quantity of droplets generated by aerosol activation, providing an aqueous medium for physical processes and chemical reactions (McNeill et al., 2012; McNeill, 2015). Within clouds, aerosol particles may undergo repeated hydration-dehydration cycles, including hygroscopic growth, activation, and subsequent evaporation. Such cloud processing could influence the concentration of OA composition (Wang et al., 2024; Gao et al., 2023; Liu et al., 2023), thereby influencing aerosol size distribution, hygroscopicity, volatility, and cloud condensation nuclei (CCN) activity (Jimenez et al., 2009; Sun et al., 2024a; Huang et al., 2018).

Additionally, cloud processing may facilitate the formation of brown carbon, including N-containing heterocyclic compounds, which could affect atmospheric radiative forcing (Liu et al., 2023)."

    *(ii)*Key uncertainties in previous studies, and the necessity of the online molecular-level approach (EESI-TOF) to address these gaps are illustrated in line 64-89 in the manuscript as follows:

    "A number of field campaigns have been conducted to measure the chemical composition of OA in cloud droplets. Several previous field campaigns found that more highly oxygenated OA is present in cloud droplets compared to cloud-free aerosol particles using online techniques, Aerosol Mass Spectrometer (AMS) or Aerodyne Aerosol Chemical Speciation Monitor (ACSM), which provide information on fragment ions of compounds, such as the fraction of m/z 44 ($CO_2^+$) in the mass spectra (Dadashazar et al., 2022; Lance et al., 2020; Gao et al., 2023). Although these studies provide valuable information on the chemical composition of aqSOA, the use of AMS or ACSM leads to molecular fragmentation and thus cannot provide molecular formulas for the components of aqSOA. As a result, the molecular composition of aqSOA and mechanisms of its formation and transformation remain incompletely understood. This gap hinders the analysis of sources, evolution, health effects, and climate impacts with respect to specific OA compounds.

    Molecular formulas of OA in cloud samples can be assigned and classified into several groups, including CHO, CHON, CHOS, and CHONS, with CHO and CHON accounting for the largest fractions (Liu et al., 2023b; Cook et al., 2017; Pailler et al., 2024; Zhao et al., 2013; Bianco et al., 2019; Sun et al., 2021; Gramlich et al., 2023). Oligomers (Cook et al., 2017; Zhao et al., 2013), organosulfates (Sun et al., 2021; Bianco et al., 2019), and N-containing compounds such as nitroaromatics (Sun et al., 2021; Cook et al., 2017; Bianco et al., 2019) have been observed in cloud droplets. Although the molecular composition of OA in cloud droplets has been characterized using offline techniques such as Fourier Transform Ion Cyclotron Resonance Mass Spectrometry (FT-ICR-MS), the formation mechanisms of many compounds in clouds remain uncertain. For example, it is not clear whether the oligomers originate from cloud processing or from aqueous aerosol due to a lack of concomitant aerosol measurements and limited temporal variation analyses (Cook et al., 2017; Zhao et al., 2013). The coarse time resolution of filter-based sampling (several hours to one day), together with limited sample numbers, prevents these studies from resolving cloud-processing reaction processes that occur on minute-to-hour timescales and are subjected to the influence of rapid variability in meteorological conditions within clouds. The chemical characteristics of aqSOA obtained from comparisons between cloud droplets and cloud-free aerosol particles are subject to large uncertainties, because the composition of both cloud droplets and aerosols may change over long-time sampling. Therefore, it is necessary to obtain online molecular information on OA in clouds by comparing OA composition of cloud droplets, interstitial aerosol particles, and cloud-free aerosol particles, to provide new insights into the detailed chemical composition, evolution variation, and the mechanism of cloud processing."

*2 . Line 246-249 and AqSOA tracers about siloxanes. The current discussion and explanation is not accurate. In particular, the concepts of "tracers" and "enhanced compounds" should be clearly distinguished. Please clarify the likely sources and implications of the siloxanes, or restructure this section accordingly. In fact, siloxanes are commonly used in personal care products and industrial chemical products. The lifetimes of the siloxanes are relatively long and will undergo atmospheric oxidation process to generate SOA, as confirmed in both laboratory and ambient studies [1, 2]. But in the aerosol phase, the high-molecular-weight siloxanes have also been detected from vehicle and jet emissions [3, 4]. The siloxanes could also be attributed to the use of silicone tubing in the sampling line [5]. Could the authors confirm that the silicone tubing was not used in the sampling system? It would also be helpful to show the time series of these compounds over the entire campaign. Was there an AMS or PTR-MS available to provide supporting information? Were any siloxane SOA tracers detected (e.g., Si- containing compounds where one or more methyl groups are replaced by hydroxyl groups)? In summary, the discussion could be better connected the focus of this study, as the siloxanes are good surfactants and could influence cloud droplet activation.*

**Response:** Thanks for the reviewer's valuable suggestion. We address the point on "tracers" and "enhanced compounds" in the next response. Regarding siloxanes, we had considered the potential contamination from semi-volatile organic compounds (SVOCs), especially siloxanes, before the campaign. Therefore, stainless steel tubing was primarily used in the sampling line, with copper tubing applied in some sections, and no silicone tubing was used.

As noted by the reviewer, sources of siloxanes are diverse and require further investigation. Discussions on siloxanes are beyond the scope of the present study and will be addressed in a separate manuscript. Consequently, to avoid confusion, we decided to exclude the siloxanes discussion in the revised manuscript, and data points of siloxanes have been removed from Figure 5 and Figure S4 (both shown below).

We have revised line 276-277 in the manuscript as follows:

"In addition, some siloxane compounds showed higher fractions in CD than in CF, such as C16H48O8Si8 (m/z 615) and C18H54O9Si9 (m/z 689) in CE2 and C12H36O6Si6 (m/z 467) and C18H54O9Si9 in CE3. Siloxane is a type of volatile chemical products such as those found in personal care products. To our best knowledge, this is the first time that C16H48O8Si8 was observed in cloud droplets. The reason for the higher fraction of siloxane warrants further study.

[Figure]

**Figure 5. Kendrick mass defect plots based on CH₂ of compounds in (a) CE2 and (b) CE3. Data points are color-coded by differences in fractions of compounds between CD and CF. Fractions of compounds are normalized to the sum of signals of all organics in CD and CF, respectively. Note that, for conciseness, data points in CE3 with normalized signal difference between −0.0003 and 0.0003**

**(appearing nearly white) are not shown here. Siloxane compounds are not shown here for clarity.**

[Figure]

**Figure S4. Kendrick mass defect plots based on O of compounds in (a) CE2 and (b) CE3. Data points are color-coded by differences in fraction of compounds between CD and CF. Fraction of compounds are normalized to the sum of signals of all organics in CD and CF, respectively. Note that, for conciseness, data points in CE3 with normalized signal difference between −0.0003 and 0.0003**

**(appearing nearly white) are not shown here. Siloxane compounds are not shown here for clarity.**

. *Section 3.3, considering the AqSOA tracers: the mass resolution and capbility of EESI-TOF are not enough to identify the structure of compounds, especially the N-containing species. Many of the proposed tracers could also arise from other sources (e.g., biomass burning, anthropogenic emissions, or in-spray artifacts) rather than exclusively as the tracer of AqSOA. For example, C9H18N2 and C12H23N are more detehced by EESI-ToF from traffic, microplastic, and agriculture, rather than the AqSOA. So it's hard to treat proposed tracers as candidates unless supported by high-resolution MS/MS, authentic standards, or orthogonal constraints (e.g., gas-phase precursors, isotopic patterns). Again, the concept of the tracers cannot be used in this kind of discussion.*

**Response:** Thanks for the reviewer's suggestion. We agree with the reviewer that the terms "aqSOA tracers" we used in the manuscript may cause some misunderstanding. The "39 tracers" in the original manuscript are compounds that are significantly enriched in cloud droplets rather than in cloud-free aerosol particles. In the revised manuscript, we change the terminology to "enriched OA compounds" which is also used in Sun et al. (2024b). The 39 enriched OA compounds are divided into four classes: CHO, CHON, CHN, and CHOSi. The CHN-class OA compounds may be heterocyclic compounds containing imine or amine functional groups, potentially resulting from secondary formation in the aqueous phase. Also, the CHN-class compounds may come from primary sources, arising from the uptake of aerosol-phase organics. $C_9H_{18}N_2$ could be emitted from traditional biomass fuel burning (Fleming et al., 2018) and $C_{12}H_{23}N$ from vehicle emissions (Thomas et al., 2025). And agricultural residue burning could emit $C_9H_{18}N_2$ and $C_{12}H_{23}N$ (Lin et al., 2012). Sources of CHOSi-class compounds warrant further investigation. The CHO, CHON, and CHN-class OA compounds could be potential aqSOA tracers for enrichment in cloud droplets. And molecular structures are warranted in future to verify formation mechanisms.

We have revised line 285-296 in the manuscript as follows:

"We identified aqSOA tracers in cloud droplets by comparing the intensity fractions of all compounds between CD and CF using a *t*-test at a significant level of 0.05. A total of 144, 421, 274, and 537 organic compounds in CE1, CE2, CE3, and CE4, respectively, passed the *t*-test. Among these compounds, 39 organic compounds in CD were significantly enriched in three or four CEs, as shown in Table 2. Two were consistently significant in CD across all four CEs: $C_{14}H_{42}O_7Si_7$ and $C_9H_{22}N_2O_4$. Furthermore, sulfate compounds were enriched in CD compared with CF in three CEs, of which time series is shown in Fig. S6. Sulfate is a well-established tracer for aqueous-phase processing, and its elevated concentration in cloud droplets and fog has been widely reported (Dadashazar et al., 2022; Brege et al., 2018; Kim et al., 2019), which further enhances the potential of identifying the enriched OA compounds as aqSOA tracers formed via cloud processing. The number of CHO, CHON, CHN, and CHOSi compounds is 15, 19, 2, and 3, respectively. The majority of the enriched OA compounds exhibit carbon numbers greater than nine, which is also an indication of accretion reactions in cloud droplets. Most of these enriched

OA compounds have not been reported in previous literature (Cook et al., 2017; Bianco et al., 2019; Sun et al., 2024b; Tong et al., 2021)."

Line 31-32 in Abstract have been revised as follows:

"We identified 39 significantly enriched compounds in CD compared with CF, which could be potentially used as aqSOA tracers formed via cloud processing."

Line 361-363 in Conclusion have been revised as follows:

"We identified several compounds significantly enriched in cloud droplets, including typical aqSOA tracers such as oxalic acid. The new aqSOA tracers, such as $C_6H_{12}O_6$ and $C_9H_{22}N_2O_4$, could help future studies identify cloud processing aqSOA."

*4 . The manuscript relies heavily on earlier and in-campaign citations (71 % before 2020) and does not sufficiently reflect recent literature (29% from the most recent 5 years, e.g. Line 48-57, line 219-222, line 246-294). In particular, line 269 lists tracer studies from 2017 and 2019, but more up-to-date work is available from 2019-2025. Note that EESI-TOF was developed around 2017, first field measurements appeared in 2018–2019 (Switzerland), and the first report of aqSOA measured by EESI was published in 2021. The manuscript should be revised to incorporate these and subsequent studies, ensuring comprehensive and current coverage beyond the authors' own campaigns.*

**Response:** Thanks for your kind suggestion. We added more references of aqSOA measured by EESI-ToF-MS in the revised manuscript as follows:

Line 47-63 in Introduction:

"Mounting evidence for aqueous secondary organic aerosol (aqSOA) has been reported in field observations in various atmospheric aqueous systems, i.e., aerosol liquid water (ALW), fog water, and cloud water. For example, several studies on source apportionment in different sites showed that aqSOA formed in ALW is an important contributor to SOA, with its fraction particularly elevated (up to 44 %) under high relative humidity (RH) conditions (Wang et al., 2021; Zhao et al., 2019; Tong et al., 2021; Gilardoni et al., 2016; Duan et al., 2022; Xu et al., 2019; Sun et al., 2016). Relative to ALW, fog water and cloud water are diluted aqueous systems where aqSOA can also be formed (Herckes et al., 2013). For fog water, the ratio of aqSOA to OA during fog-rain days is enhanced compared with non-fog-rain days (Duan et al., 2021). Additionally, OA composition of fog water is more oxidized (Brege et al., 2018), has more N-containing compounds (Mattsson et al., 2025; Sun et al., 2024a; Kim et al., 2019) compared with aerosol particles, and shows signs of oligomerization based on fragments in the mass spectrum (Gilardoni et al., 2016; Mandariya et al., 2019). In contrast to fog, the cloud is more common, ubiquitously presents in the atmosphere, and consists of a large quantity of droplets generated by aerosol activation, providing an aqueous medium for physical processes and chemical reactions (McNeill et al., 2012; McNeill, 2015). Within clouds, aerosol particles may undergo repeated hydration-dehydration cycles, including hygroscopic growth, activation, and subsequent evaporation. Such cloud processing could influence the concentration of OA composition (Wang et al., 2024b; Gao et al., 2023; Liu et al., 2023b), thereby influencing aerosol size distribution, hygroscopicity, volatility, and cloud condensation nuclei (CCN) activity (Jimenez et al., 2009). Additionally, cloud processing may facilitate the formation of brown carbon, including N-containing heterocyclic compounds, which could affect atmospheric radiative forcing (Liu et al., 2023b)."

Line 120-121 in Methods:

"Detailed information regarding EESI-ToF-MS has been reported previously (Lopez-Hilfiker et al., 2019; Stefenelli et al., 2019; Brown et al., 2021; Kumar et al., 2022; Luo et al., 2024; Xue et al., 2025)."

Line 238-248 in Sect. 3.2:

"Collectively, compounds in CD had more $n_C$, $n_O$, and $n_N$ than those in CF. The molecular characteristic of higher $n_C$ is likely attributed to accretion reactions such as oligomerization (Yu et al., 2016; Fenselau et al., 2025). This finding is consistent with several laboratory studies of aqSOA formation. For instance, enriched high-molecular-weight compounds (HMWC) in aqSOA were reported in the bulk phase experiments of methylglyoxal and glyoxal under cloud-relevant conditions (Tan et al., 2009; Altieri et al., 2008). And aqSOA from in-cloud simulation using a wetted-wall flow reactor has more highly oxygenated and carbon-containing compounds than gasSOA simulated by an oxidation flow reactor (OFR) from the same biomass burning samples (Wang et al., 2024b). Experiments in the bulk phase and the wetted-wall flow reactor which better represents atmospheric aqueous conditions, indicate that accretion reactions could be prevalent in cloud droplets. Field observations in the Arctic also show potential evidence of accretion reactions, with compounds of longer carbon chains enriched in CD relative to CF (Pasquier et al., 2022), hinting at the possible importance of accretion reactions. Notably, this study provides direct molecular-level evidence for the contribution of accretion reactions during cloud processing of OA."

Line 292-296 in Sect. 3.3:

"The number of CHO, CHON, CHN, and CHOSi compounds is 15, 19, 2, and 3, respectively. The majority of the enriched OA compounds exhibit carbon numbers greater than nine, which is also an indication of accretion reactions in cloud droplets. Most of these enriched OA compounds have not been reported in previous literature (Cook et al., 2017; Bianco et al., 2019;

Tong et al., 2021; Sun et al., 2024b).

*Minor comments:*

*Line 61: "Aerosol Mass Spectrometer (AMS) or Aerodyne Aerosol Chemical Speciation Monitor (ACSM)".*

**Response:** We thank the reviewer for the kind remarks. We have revised line 64-68 in the manuscript as follows:

"A number of field campaigns have been conducted to measure chemical composition of OA in cloud droplets. Several previous field campaigns found that more highly oxygenated OA was present in cloud droplets compared to cloud-free aerosol particles using online techniques, Aerosol Mass Spectrometer (AMS) or Aerodyne Aerosol Chemical Speciation Monitor (ACSM), which obtained information on fragment ions of compounds, such as the fraction of m/z 44 ($CO_2^+$) in the mass spectra (Dadashazar et al., 2022; Lance et al., 2020; Gao et al., 2023)."

*Line 82: revise to "...at the summit of Damaojian Mountain, located in Jinhua City, Zhejiang Province, China."*

**Response:** We have revised line 97-99 in the manuscript as follows:

    "We conducted this field campaign from May 1$^{st}$ to May 29$^{th}$ in 2024 at Shanghuang Eco-Environmental Observatory of Chinese Academy of Sciences at the summit of the Damaojian mountain (119.51° E and 28.58° N, 1128 m above sea level)

located in Jinhua city, Zhejiang province, China."

*Line 105: delete "Here is a brief introduction.", too informal.*

**Response:** We have revised line 119-124 in the manuscript as follows:

    "Detailed information regarding EESI-ToF-MS has been reported previously (Lopez-Hilfiker et al., 2019; Stefenelli et al., 2019; Brown et al., 2021; Kumar et al., 2022; Luo et al., 2024; Xue et al., 2025).  Aerosol was sampled after gaseous compounds were removed by entering a charcoal denuder, and subsequently intersected with an electrospray generated from a working solution containing 100 ppm NaI in a 1:1 (v/v) water and acetonitrile mixture, allowing aerosol compounds to be detected as [M+Na]$^+$ in positive ion mode."

*Line 115: "signal-to-background ration (s/b)".*

**Response:** We have revised line 129-132 in the manuscript as follows:

    "Mass spectral data were processed using Tofware 3.2.5 in Igor Pro 8. For data screening, the signal-to-background ratio (s/b) was calculated as the median value of (sample signal−background)/background, thereby excluding compounds showing insignificant differences between sample and background."

*Line 268: "To the best of our knowledge, this is the first observation of $C16H48O8Si8$ ...". As mentioned in major comments 2, the description needs to be considered.*

**Response:** We have deleted this sentence based on the response to major comment#2 in the revised manuscript.

*Line 308: hours to a day.*

**Response:** This has been modified in line 343-345 in the revised manuscript as follows:

"Most previous comparisons of the chemical composition of cloud droplets with cloud-free aerosol particles or interstitial aerosol particles are based on long sampling (hours to a day) and offline analysis (Sun et al., 2021; Liu et al., 2023b)."

1. *Chen, Y., et al., Chemical characterization and formation of secondary organosiloxane aerosol (SOSiA) from OH oxidation of decamethylcyclopentasiloxane. Environmental Science: Atmospheres, 2023. 3(4): p. 662-671.*

2. *Meepage, J.N., et al., Advances in the Separation and Detection of Secondary Organic Aerosol Produced by Decamethylcyclopentasiloxane (D5) in Laboratory-Generated and Ambient Aerosol. ACS ES&T Air, 2024. 1(5): p.*
*365-375.*

3. *Yao, P., et al., Methylsiloxanes from Vehicle Emissions Detected in Aerosol Particles. Environmental Science & Technology, 2023. 57(38): p. 14269-14279.*

4. *Decker, Z.C.J., et al., Emission and Formation of Aircraft Engine Oil Ultrafine Particles. ACS ES&T Air, 2024. 1(12): p. 1662-1672.*

5. *Timko, M.T., et al., Composition and Sources of the Organic Particle Emissions from Aircraft Engines. Aerosol Science and Technology, 2014. 48(1): p. 61-73.*

---

## Author Comment (AC2)

**Responses to all reviewers:**

We thank the reviewers for the careful review of our manuscript. The comments and suggestions are greatly appreciated and have helped to further improve the manuscript. As shown below, all the comments have been addressed. In the following, please find our responses to the comments and the corresponding revisions made to the manuscript. The point-by-point comments are addressed in the following text. The original comments are shown in italics and responses are in normal font. The revised parts of the manuscript are in blue. The line numbers here refer to the clean version.

**Reviewer: 2**

*The study characterized the molecular composition of the organic fraction of cloud droplet residue, pre-cloud aerosols, interstitial aerosols, and post-cloud aerosols using EESI-ToF-MS. The approach yielded a high-resolution dataset that captured detailed temporal variations in molecular composition. While the study provided additional evidence on the relative contributions of CHO and CHON species in cloud droplet residues and aerosols, its findings largely corroborated previous results rather than extending current understanding. The authors interpreted their data mostly relying on findings from previous studies.*

**Response:** We thank the reviewer for the comments on our manuscript. While we appreciate the reviewer's perspective, we respectfully disagree with the comment that our "findings largely corroborated previous results rather than extending current understanding". Here are three new findings in our study that extend previous understanding.

1. We find that the CHON compounds enhanced in cloud droplets compared to interstitial and cloud-free aerosol particles are mostly contributed by compounds with an O/N ratio of ≥3, likely organonitrates, amino acids, or nitrogen-heterocyclic compounds. This finding is in contrast to previous studies. E.g., Liu et al. (2023) reported that the number of CHON compounds was higher in cloud droplets in Mt. Tai using FT-ICR-MS, and more CHON compounds had reduced nitrogen groups (O/N <3). Our results indicate different formation mechanisms of CHON compounds.

2. We provide direct molecular-level evidence for the contribution of accretion reactions during cloud processing of OA. Fog samples measured by AMS were compared with aerosol at different stages of fog episodes, suggesting that oligomer-ization could be significant in fog processing of OA (Mandariya et al., 2019). However, due to the fragmentation in AMS, there was no direct evidence. Although cloud observations using FT-ICR-MS have reported the presence of oligomers in cloud samples, such studies could not determine the source of oligomers, whether they originated from cloud processing or from uptake of gas-phase or aqueous aerosol, because no concomitant aerosol samples were collected for comparison (Zhao et al., 2013; Cook et al., 2017). In this study, by comparing OA composition in cloud droplets with that in cloud-free aerosol particles, we provide molecular-level evidence from cloud observations and prove that accretion reactions

30        have a substantial contribution to aqSOA.

3.    We find several enriched compounds in cloud droplets compared to cloud-free aerosol particles. Most of the enriched OA compounds have not been reported in previous literature (Cook et al., 2017; Bianco et al., 2019; Tong et al., 2021; Sun et al., 2024b), and could possibly be used as tracers of cloud-processing aqSOA. This can be helpful for further study to identify the contribution of cloud processing.

In addition, previous offline studies are limited by low temporal resolution and limited sample numbers (Sun et al., 2021; Cook et al., 2017; Pailler et al., 2024), whereas existing online measurements provided only fragmented OA information without molecular formulas (Dadashazar et al., 2022; Lance et al., 2020). By combining 20-s time resolution with molecular-level characterization from EESI-ToF-MS, this study tracks the formation and evolution of aqSOA during cloud processing and offers new insights into cloud-influenced organic aerosol chemistry.

We thank the reviewer for the comments. We realize that some descriptions in the original manuscript were not sufficiently clear to convey our intended findings, which may have led to misunderstanding. To address this, we have strengthened the explanation of the novelty and clarified the related statements in the introduction, discussion, and conclusion of the revised manuscript as follows:

Line 64-89 in Introduction:

"A number of field campaigns have been conducted to measure the chemical composition of OA in cloud droplets. Several previous field campaigns found that more highly oxygenated OA is present in cloud droplets compared to cloud-free aerosol particles using online techniques, Aerosol Mass Spectrometer (AMS) or Aerodyne Aerosol Chemical Speciation Monitor (ACSM), which provide information on fragment ions of compounds, such as the fraction of m/z 44 ($CO_2^+$) in the mass spectra (Dadashazar et al., 2022; Lance et al., 2020; Gao et al., 2023). Although these studies provide valuable information on the chemical composition of aqSOA, the use of AMS or ACSM leads to molecular fragmentation and thus cannot provide molecular formulas for the components of aqSOA. As a result, the molecular composition of aqSOA and mechanisms of its formation and transformation remain incompletely understood. This gap hinders the analysis of sources, evolution, health effects, and climate impacts with respect to specific OA compounds.

Molecular formulas of OA in cloud samples can be assigned and classified into several groups, including CHO, CHON, CHOS, and CHONS, with CHO and CHON accounting for the largest fractions (Liu et al., 2023; Cook et al., 2017; Pailler et al., 2024; Zhao et al., 2013; Bianco et al., 2019; Sun et al., 2021; Gramlich et al., 2023). Oligomers (Cook et al., 2017; Zhao et al., 2013), organosulfates (Sun et al., 2021; Bianco et al., 2019), and N-containing compounds such as nitroaromatics (Sun et al., 2021; Cook et al., 2017; Bianco et al., 2019) have been observed in cloud droplets. Although the molecular composition of OA in cloud droplets has been characterized using offline techniques such as Fourier Transform Ion Cyclotron Resonance

60    Mass Spectrometry (FT-ICR-MS), the formation mechanisms of many compounds in clouds remain uncertain. For example, it is not clear whether the oligomers originate from cloud processing or from aqueous aerosol due to a lack of concomitant aerosol measurements and limited temporal variation analyses (Cook et al., 2017; Zhao et al., 2013). The coarse time resolution of filter-based sampling (several hours to one day), together with limited sample numbers, prevents these studies from resolving cloud-processing reaction processes that occur on minute-to-hour timescales and are subjected to the influence of

65    rapid variability in meteorological conditions within clouds. The chemical characteristics of aqSOA obtained from comparisons between cloud droplets and cloud-free aerosol particles are subject to large uncertainties, because the composition of both cloud droplets and aerosols may change over long-time sampling. Therefore, it is necessary to obtain online molecular information on OA in clouds by comparing OA composition of cloud droplets, interstitial aerosol particles, and cloud-free aerosol particles, to provide new insights into the detailed chemical composition, evolution variation, and the mechanism of

70    cloud processing."

Line 245-248 in Sect 3.2:

"Field observations in the Arctic also show potential evidence of accretion reactions, with compounds of longer carbon chains enriched in CD relative to CF (Pasquier et al., 2022), hinting at the possible importance of accretion reactions. Notably, this study provides direct molecular-level evidence for the contribution of accretion reactions during cloud processing of OA."

75    Line 352-389 in Conclusion:

"AqSOA molecular composition and processing in cloud episodes were studied using online molecular information obtained by EESI-ToF-MS at a high-mountain site in China. Cloud processing substantially influences OA composition, resulting in large differences among distinct cloud episodes. Organics in cloud droplets had an average molecular formula $C_{9.95–12.92}H_{14.53–21.78}O_{5.15–6.02}N_{0.32–0.42}S_{0–0.01}Si_{0–1.29}$ for the selected four cloud episodes. CHO compounds contributed

80    predominantly to OA in cloud droplets. CHON was enhanced markedly in cloud droplets compared with cloud-free aerosol particles and interstitial aerosol particles in most cloud episodes. The majority of CHON compounds were likely organonitrates, highlighting the enrichment of organonitrates compounds in cloud processing. OA in cloud droplets contained higher numbers of C, O, and N atoms, exhibited a $CH_2$-based homologous series, and showed an enrichment of higher-molecular-weight compounds compared with adjacent cloud-free aerosol particles, collectively highlighting the importance of accretion reactions

85    in cloud processing of OA at the molecular level. We identified several compounds significantly enriched in cloud droplets, including typical aqSOA tracers such as oxalic acid. The new aqSOA tracers, such as $C_6H_{12}O_6$ and $C_9H_{22}N_2O_4$, could help future studies identify cloud processing aqSOA.

This study provides direct molecular-level evidence for the contribution of accretion reactions during cloud processing of OA. Although previous cloud observations using FT-ICR-MS reported the presence of oligomers in cloud samples, these

studies could not distinguish whether such compounds originated from cloud processing or aqueous aerosols, as no concomitant aerosol samples were collected for comparison (Zhao et al., 2013; Cook et al., 2017). By directly comparing OA composition in cloud droplets with that in cloud-free aerosol particles, our results clearly demonstrate that accretion reactions occur within cloud droplets. It has been assumed that HMWC are predominantly formed in aerosol liquid water rather than cloud water, owing to the lower reaction rates of accretion reactions in the more dilute cloud-water environment (Ervens et al., 2011). In contrast, our study provides direct molecular-level evidence that such compounds can also be formed in cloud water, extending earlier observations by Cook et al. (2017). These findings highlight that accretion reactions should be considered when modeling aqSOA formation in clouds.

The HMWC formed via accretion reaction may have implications for the environment and climate. Due to the increase in the HMWC, accretion reactions likely reduce the volatility of organics and could potentially enhance OA mass concentration and alter the aerosol size distribution after cloud evaporation. The formation of HMWC can also modify physicochemical properties, such as lifetime, oxidation state, viscosity, and hygroscopic properties, which may further influence the cloud activation of these aerosols. In addition, the formation of N-containing compounds in cloud droplets, such as organonitrates, pyrrole, and imidazole, may also affect the physicochemical properties of aqSOA, e.g., contributing to brown carbon and thus affecting regional radiative forcing.

Based on the measurement of high time resolution (~20 s), we find that the concentrations of individual organic compounds were highly dynamic in clouds, which is likely due to the turbulence in clouds. Such a highly dynamic nature in clouds poses difficulties in extracting the influence of chemical processes on individual compounds for instrumentation with low temporal resolution. Therefore, our results highlight the necessity of high time resolution measurements (< 1 h), especially online systems achieving minute-level resolution to investigate the chemical processes in clouds, considering dynamic variations of compounds in clouds due to turbulence in clouds and alterations in air masses.

It should be noted that this study provides molecular formulas only, while detailed structural information is warranted to better constrain the sources, formation mechanisms, and climate impacts of aqSOA in clouds. In addition, sources of compounds enriched in cloud droplets will be investigated in future studies."

*The study's main points were: 1. 39 new compounds in cloud droplet residues; and 2. the hypothesis that accretion reactions were promoted during cloud processing. However, it remains unclear why these new 39 compounds were detected. Were they revealed because of the dataset's high temporal resolution, the unique analytical technique, or chemistry specific to the sampled environment?*

**Response:** The "39 compounds" enriched in cloud droplets were detected owing to the high temporal resolution of our measurements and the molecular-level comparison between cloud droplets and cloud-free aerosol particles. The enhancement of OA compounds in cloud droplets could be blurred by methods with low temporal resolution. Additionally, the observation site is subjected to diverse influences, including both anthropogenic and biogenic contributions, where aqueous processing of various compounds can be identified. We would like to clarify that the "39 new compounds" refer to newly identified aqSOA tracers that expand the existing reservoir of known tracers. In fact, some of these compounds have been reported in previous aerosol studies (e.g., $C_{10}H_{16}O_2$ in Qi et al. (2019)), but their enrichment in cloud droplets and their role as aqSOA tracers have not been recognized before.

And following Reviewer 1's suggestion, we have modified the discussion of the 39 enriched compounds in line 285-296 in Sect. 3.3 of the revised manuscript to clarify their definition as follows:

"We identified aqSOA tracers in cloud droplets by comparing the intensity fractions of all compounds between CD and CF using a $t$-test at a significant level of 0.05. A total of 144, 421, 274, and 537 organic compounds in CE1, CE2, CE3, and CE4, respectively, passed the $t$-test. Among these compounds, 39 organic compounds in CD were significantly enriched in three or four CEs, as shown in Table 2. Two were consistently significant in CD across all four CEs: $C_{14}H_{42}O_7Si_7$ and $C_9H_{22}N_2O_4$. Furthermore, sulfate compounds were enriched in CD compared with CF in three CEs, of which time series is shown in Fig. S6. Sulfate is a well-established tracer for aqueous-phase processing, and its elevated concentration in cloud droplets and fog has been widely reported (Dadashazar et al., 2022; Brege et al., 2018; Kim et al., 2019), which further enhances the potential of identifying the enriched OA compounds as aqSOA tracers formed via cloud processing. The number of CHO, CHON, CHN, and CHOSi compounds is 15, 19, 2, and 3, respectively. The majority of the enriched OA compounds exhibit carbon numbers greater than nine, which is also an indication of accretion reactions in cloud droplets. Most of these enriched OA compounds have not been reported in previous literature (Cook et al., 2017; Bianco et al., 2019; Sun et al., 2024; Tong et al., 2021)."

*The study period experienced four cloud events, effectively providing four data points for comparing pre-cloud, CD, INT and CF samples. Caution should be exercised in drawing general conclusions from such a limited dataset. For instance, CE2 was an exception in which CHON did not comprise the largest fraction of total OA, and CHO was not the lowest among the four sample types. Thus, one quarter of the samples did not align with the generalization, and it would be advisable to moderate some of the claims derived from these comparisons.*

**Response:** Thanks for the reviewer's reminder and comments. CE2 shows an exception of CHO and CHON characteristics from other CEs due to air mass changes during the long-time interval between post-cloud and others (pre-cloud, CD, and INT).

To make the statements more accurate, we have revised line 188-191 in the manuscript as follows:

"In most cloud episodes, CD showed the lowest CHO fraction (54.8 %–70.7 %) and the highest CHON fraction (26.6 %–33.2 %) among the four sample types. In CE2, with higher CHON (29.4 %) and lower CHO (54.6 %) in post-cloud aerosols than other sample types, is attributable to air mass changes during the long-time interval between post-cloud and others (shown in Fig. 2 and Fig. S1), as indicated by elevated CO concentration."

Line 28-30 in Abstract are revised as follows:

"In most cloud episodes, the fraction of CHO was lower in CD than that in INT and CF, while the fraction of CHON was higher, which may result from the uptake of organonitrates or nitration in cloud water."

Line 356-357 in Conclusion are revised as follows:

"CHON was enhanced markedly in cloud droplets compared with cloud-free aerosol particles and interstitial aerosol particles in most cloud episodes."

*Specific comments:*

1. *Line 27: "With adjacent time" is not clear. Adjacent to what?*

**Response:** We intended to express that the cloud-free aerosol particles were sampled immediately before or after each in-cloud stage, ensuring that the comparison among cloud droplets, interstitial aerosol particles, and cloud-free aerosol particles was conducted under similar meteorological and emission conditions. Therefore, the differences observed in OA composition can be attributed primarily to cloud processing rather than temporal variability. To avoid ambiguity, the phrase "with adjacent time" has been removed, and the expression has been revised for clarity in the manuscript.

Line 26-28 in the revised manuscript have been modified as follows:

"We identified 2084 molecular formulas and compared OA composition from three sample types : cloud droplets (CD), interstitial aerosol particles (INT), and cloud-free aerosol particles (CF) in representative cloud episodes."

2. *Line 44-46: Research on the role of aqueous-phase chemistry in SOAs has been going on for more than four decades. Are the studies cited here seminal ones?*

**Response:** Thank you for the suggestion. We have gone through the literature on the role of aqueous-phase chemistry in SOA and added references in line 45-46 in Introduction as follows:

"In addition to the traditional gas-phase processing, aqueous-phase pathways have been recognized as an important source of SOA (Sehested et al., 1975; Galloway et al., 1976; Graedel and Weschler, 1981; Fu et al., 2008; Tan et al., 2009; Zhang et al., 2010; Ervens et al., 2011; Lamkaddam et al., 2021)."

3. *Line 51-52: What are the main findings from Duan et al. (2021)?*

**Response:** Thanks for the reminder. We have expanded the main findings of Duan et al. (2021) in Line 52-55 in the revised manuscript as follows:

"For fog water, the ratio of aqSOA to OA during fog-rain days is enhanced compared with non-fog-rain days (Duan et al., 2021). Additionally, OA composition of fog water is more oxidized (Brege et al., 2018), has more N-containing compounds (Mattsson et al., 2025; Sun et al., 2024a; Kim et al., 2019) compared with aerosol particles, and shows signs of oligomerization based on fragments in the mass spectrum (Gilardoni et al., 2016; Mandariya et al., 2019)."

4. *Lines 151-153: Are the ranges for C, H, O, and N – C:10.01–12.81 vs 8.43–11.10, H: 14.59–20.34 vs. 14.23–16.83, O: 5.08–6.00 vs. 5.06–5.72, and N: 0.34–0.43 vs. 0.16–0.35 – between CD and pre-loud really that different?*

**Response:** We have done statistics tests, which show that the numbers of C, H, O, and N atoms of organic compounds in CD are indeed significantly higher compared to pre-cloud ($p < 0.05$). In addition, we have updated these values according to an optimized method in which raw signal values at each time in CD and pre-cloud are used to calculate the average molecular formula, while average signal values in the whole CD and pre-cloud periods were used in the original manuscript. The increase in $n_C$ suggests the occurrence of accretion reactions in cloud droplets, leading to the formation of higher-molecular-weight compounds. The enhancement in $n_O$ indicates a higher degree of oxidation in cloud droplets. The increase in $n_N$ is consistent with the results discussed in Sect. 3.2, where we demonstrated that CHON compounds are enriched in cloud droplets relative to cloud-free aerosol particles.

Line 167-178 in Sect. 3.1 are revised as follows:

"A total of 2084 molecular formulas of OA were identified in the campaign. Mean formula of CD was $C_{9.95–12.92}H_{14.53–21.78}O_{5.15–6.02}N_{0.32–0.42}S_{0–0.01}Si_{0–1.29}$ for CE1–CE4. Compared with pre-cloud aerosols with formula $C_{8.17–10.57}H_{12.99–16.14}O_{4.98–5.64}N_{0.12–0.30}S_{0–0.01}Si_{0–0.24}$, CD exhibited increased numbers of carbon, hydrogen, oxygen, and nitrogen atoms, with the differences being statistically significant ($p < 0.05$) (Table S2). These molecular formulas were classified into eight classes, that is, CHO (only C, H, O atoms are contained in the chemical formula, hereafter), CHON, CHONS, CHOS, CHN, CHS, CHNS, and CHOSi. Since the composition of OA varied in different CEs, the fractions of these OA classes are discussed for each CE in Sect. 3.2. The O/C ratio was generally lower in CD (0.45–0.66) than in pre-cloud aerosols, INT, and post-cloud

aerosols in 4 CEs. The O/C ratio in CD is comparable to those reported for fog water (0.52–0.68), aqSOA (0.61–0.84), and oxygenated OA (0.44–0.83) by Gilardoni et al. (2016). In general, O/C of CD in this study is comparable to that of fog (0.58–0.8) in the Po Valley in Brege et al. (2018), while H/C of CD (1.50–1.87) is higher than that of fog (1.29–1.37) in that study. Furthermore, CD showed elevated N/C (0.033–0.045) relative to other sample types, while its OSc value (−0.83 to −0.25) is generally lower than in other sample types."

We have added the average atom numbers of C, H, O, and N with standard deviation in Table S2 in the revised supplementary as follows:

**Table S2: Mean atom numbers ($n_C$, $n_H$, $n_O$, and $n_N$; mean ± standard deviation) of OA for CE1–CE4 in pre-cloud aerosols, CD, INT, and post-cloud aerosols.**

| | | Pre-cloud aerosols | CD | INT | Post-cloud aerosols |
|-----|-----|------------|------------|------------|------------|
| CE1 | $n_C$ | 8.72±0.38 | 10.45±1.07 | 8.99±0.53 | 8.86±0.41 |
| | $n_H$ | 13.81±1.09 | 17.25±3.16 | 14.21±1.17 | 14.45±1.10 |
| | $n_O$ | 5.64±0.12 | 6.02±0.33 | 5.68±0.13 | 5.71±0.15 |
| | $n_N$ | 0.20±0.03 | 0.32±0.08 | 0.26±0.05 | 0.22±0.04 |
| CE2 | $n_C$ | 8.33±0.20 | 10.13±1.26 | 8.52±0.34 | 11.12±0.55 |
| | $n_H$ | 13.72±0.65 | 19.51±4.50 | 14.19±1.05 | 21.10±2.80 |
| | $n_O$ | 4.98±0.12 | 5.44±0.57 | 4.92±0.18 | 6.26±0.20 |
| | $n_N$ | 0.28±0.04 | 0.39±0.09 | 0.33±0.07 | 0.33±0.07 |
| CE3 | $n_C$ | 10.57±0.18 | 12.92±0.70 | 10.42±0.09 | 10.51±0.13 |
| | $n_H$ | 16.14±0.30 | 21.78±1.76 | 16.21±0.20 | 16.18±0.29 |
| | $n_O$ | 5.08±0.05 | 5.15±0.12 | 5.09±0.03 | 5.11±0.04 |
| | $n_N$ | 0.30±0.02 | 0.42±0.06 | 0.27±0.02 | 0.27±0.02 |
| CE4 | $n_C$ | 8.17±0.15 | 9.95±0.49 | 8.94±0.20 | 8.77±0.12 |
| | $n_H$ | 12.99±0.19 | 14.53±0.65 | 13.36±0.26 | 13.20±0.17 |
| | $n_O$ | 5.29±0.08 | 5.68±0.23 | 5.52±0.08 | 5.52±0.06 |
| | $n_N$ | 0.12±0.03 | 0.32±0.08 | 0.20±0.04 | 0.18±0.02 |

Table 1 is revised as follows:

230 **Table 1. Mean molecular formulas and elemental parameters (H/C, O/C, N/C, and OSc; mean ± standard deviation) of OA for CE1–CE4 in pre-cloud aerosols, CD, INT, and post-cloud aerosols.**

| | | Pre-cloud aerosols | CD | INT | Post-cloud aerosols |
|---|---|---|---|---|---|
| CE1 | Formula | $C_{8.72}H_{13.81}O_{5.64}N_{0.20}S_0Si_{0.24}$ | $C_{10.45}H_{17.25}O_{6.02}N_{0.32}S_0Si_{0.69}$ | $C_{8.99}H_{14.21}O_{5.68}N_{0.26}S_0Si_{0.23}$ | $C_{8.86}H_{14.45}O_{5.71}N_{0.22}S_0Si_{0.40}$ |
| | H/C | 1.61±0.04 | 1.64±0.11 | 1.61±0.04 | 1.61±0.04 |
| | O/C | 0.71±0.02 | 0.66±0.03 | 0.69±0.02 | 0.70±0.02 |
| | N/C | 0.022±0.004 | 0.033±0.008 | 0.028±0.005 | 0.025±0.006 |
| | OSc | -0.19±0.06 | -0.33±0.14 | -0.22±0.06 | -0.20±0.06 |
| CE2 | Formula | $C_{8.33}H_{13.72}O_{4.98}N_{0.28}S_{0.01}Si_{0.16}$ | $C_{10.13}H_{19.51}O_{5.44}N_{0.39}S_{0.01}Si_{1.29}$ | $C_{8.52}H_{14.19}O_{4.92}N_{0.33}S_{0.01}Si_{0.20}$ | $C_{11.12}H_{21.10}O_{6.26}N_{0.33}S_{0.01}Si_{1.52}$ |
| | H/C | 1.70±0.03 | 1.87±0.15 | 1.72±0.04 | 1.78±0.13 |
| | O/C | 0.63±0.01 | 0.58±0.03 | 0.61±0.02 | 0.61±0.02 |
| | N/C | 0.037±0.004 | 0.045±0.01 | 0.042±0.009 | 0.031±0.006 |
| | OSc | -0.43±0.04 | -0.71±0.19 | -0.50±0.08 | -0.56±0.15 |
| CE3 | Formula | $C_{10.57}H_{16.14}O_{5.08}N_{0.30}S_{0.01}Si_{0.02}$ | $C_{12.92}H_{21.78}O_{5.15}N_{0.42}S_{0.01}Si_{0.87}$ | $C_{10.42}H_{16.21}O_{5.09}N_{0.27}S_{0.004}Si_{0.11}$ | $C_{10.51}H_{16.18}O_{5.11}N_{0.27}S_{0.01}Si_{0.07}$ |
| | H/C | 1.57±0.01 | 1.73±0.08 | 1.59±0.01 | 1.58±0.01 |
| | O/C | 0.55±0.008 | 0.45±0.03 | 0.55±0.005 | 0.55±0.005 |
| | N/C | 0.033±0.002 | 0.040±0.007 | 0.028±0.002 | 0.029±0.002 |
| | OSc | -0.48±0.02 | -0.83±0.12 | -0.49±0.02 | -0.47±0.02 |
| CE4 | Formula | $C_{8.17}H_{12.99}O_{5.29}N_{0.12}S_{0.004}Si_0$ | $C_{9.95}H_{14.53}O_{5.68}N_{0.32}S_{0.01}Si_0$ | $C_{8.94}H_{13.36}O_{5.52}N_{0.20}S_{0.004}Si_0$ | $C_{8.77}H_{13.20}O_{5.52}N_{0.18}S_{0.004}Si_0$ |
| | H/C | 1.65±0.02 | 1.50±0.04 | 1.52±0.02 | 1.52±0.01 |
| | O/C | 0.69±0.01 | 0.62±0.02 | 0.66±0.008 | 0.67±0.01 |
| | N/C | 0.014±0.005 | 0.034±0.01 | 0.021±0.005 | 0.019±0.003 |
| | OSc | -0.28±0.04 | -0.25±0.07 | -0.20±0.02 | -0.18±0.02 |

*5. Line 278: "Large fluctuations" in what?*

235 **Response:** We meant large inhomogeneity within clouds due to strong turbulence. We have revised the sentence in Line 305-306 as follows:

"The oxalic acid signal was exclusively observed during CD, whereas it remained weak and noisy in CF and INT, as shown in Fig. 6. The oxalic acid signal was significantly enhanced only in CE2 and CE4, rather than in all CEs, which may be related to larger inhomogeneity within clouds due to strong turbulence."

240

**References:**

Bianco, A., Riva, M., Baray, J. L., Ribeiro, M., Chaumerliac, N., George, C., Bridoux, M., and Deguillaume, L.: Chemical Characterization of Cloudwater Collected at Puy de Dome by FT-ICR MS Reveals the Presence of SOA Components, ACS Earth Space Chem., 3, 2076-2087, https://doi.org/10.1021/acsearthspacechem.9b00153, 2019.

245 Brege, M., Paglione, M., Gilardoni, S., Decesari, S., Facchini, M. C., and Mazzoleni, L. R.: Molecular insights on aging and aqueous-phase processing from ambient biomass burning emissions-influenced Po Valley fog and aerosol, Atmos. Chem. Phys., 18, 13197-13214, https://doi.org/10.5194/acp-18-13197-2018, 2018.

Cook, R. D., Lin, Y. H., Peng, Z., Boone, E., Chu, R. K., Dukett, J. E., Gunsch, M. J., Zhang, W., Tolic, N., Laskin, A., and Pratt, K. A.: Biogenic, urban, and wildfire influences on the molecular composition of dissolved organic compounds in
250 cloud water, Atmos. Chem. Phys., 17, 15167-15180, https://doi.org/10.5194/acp-17-15167-2017, 2017.

Dadashazar, H., Corral, A. F., Crosbie, E., Dmitrovic, S., Kirschler, S., McCauley, K., Moore, R., Robinson, C., Schlosser, J. S., Shook, M., Thornhill, K. L., Voigt, C., Winstead, E., Ziemba, L., and Sorooshian, A.: Organic enrichment in droplet residual particles relative to out of cloud over the northwestern Atlantic: analysis of airborne ACTIVATE data, Atmos. Chem. Phys., 22, 13897-13913, https://doi.org/10.5194/acp-22-13897-2022, 2022.

255 Ervens, B., Turpin, B. J., and Weber, R. J.: Secondary organic aerosol formation in cloud droplets and aqueous particles (aqSOA): a review of laboratory, field and model studies, Atmos. Chem. Phys., 11, 11069-11102, https://doi.org/10.5194/acp-11-11069-2011, 2011.

Fu, T.-M., Jacob, D. J., Wittrock, F., Burrows, J. P., Vrekoussis, M., and Henze, D. K.: Global budgets of atmospheric glyoxal and methylglyoxal, and implications for formation of secondary organic aerosols, J. Geophys. Res., 113, D15303,
260 https://doi.org/10.1029/2007jd009505, 2008.

Galloway, J. N., Likens, G. E., and Edgerton, E. S.: Acid Precipitation In the Northeastern United States: pH and Acidity, Science, 194, 722-724, https://doi.org/10.1126/science.194.4266.722, 1976.

Gao, M., Zhou, S., He, Y., Zhang, G., Ma, N., Li, Y., Li, F., Yang, Y., Peng, L., Zhao, J., Bi, X., Hu, W., Sun, Y., Wang, B., and Wang, X.: In Situ Observation of Multiphase Oxidation-Driven Secondary Organic Aerosol Formation during Cloud
265 Processing at a Mountain Site in Southern China, Environ. Sci. Technol. Lett., 10, 573−581, https://doi.org/10.1021/acs.estlett.3c00331, 2023.

Graedel, T. E. and Weschler, C. J.: Chemistry Within Aqueous Atmospheric Aerosols And Raindrops, Reviews of Geophysics, 19, 505-539, https://doi.org/10.1029/RG019i004p00505, 1981.

Gramlich, Y., Siegel, K., Haslett, S. L., Freitas, G., Krejci, R., Zieger, P., and Mohr, C.: Revealing the chemical characteristics
270 of Arctic low-level cloud residuals – in situ observations from a mountain site, Atmos. Chem. Phys., 23, 6813-6834, https://doi.org/10.5194/acp-23-6813-2023, 2023.

Kim, H., Collier, S., Ge, X., Xu, J., Sun, Y., Jiang, W., Wang, Y., Herckes, P., and Zhang, Q.: Chemical processing of water-soluble species and formation of secondary organic aerosol in fogs, Atmos. Environ., 200, 158-166, https://doi.org/10.1016/j.atmosenv.2018.11.062, 2019.

275 Lamkaddam, H., Dommen, J., Ranjithkumar, A., Gordon, H., Wehrle, G., Krechmer, J., Majluf, F., Salionov, D., Schmale, J., Bjelic, S., Carslaw, K. S., El Haddad, I., and Baltensperger, U.: Large contribution to secondary organic aerosol from isoprene cloud chemistry, Sci. Adv., 7, eabe2952, https://doi.org/10.1126/sciadv.abe2952, 2021.

Lance, S., Zhang, J., Schwab, J. J., Casson, P., Brandt, R. E., Fitzjarrald, D. R., Schwab, M. J., Sicker, J., Lu, C. H., Chen, S. P., Yun, J., Freedman, J. M., Shrestha, B., Min, Q. L., Beauharnois, M., Crandall, B., Joseph, E., Brewer, M. J., Minder,
280 J. R., Orlowski, D., Christiansen, A., Carlton, A. G., and Barth, M. C.: Overview of the CPOC Pilot Study at Whiteface Mountain, NY Cloud Processing of Organics within Clouds (CPOC), Bull. Amer. Meteorol. Soc., 101, E1820-E1841, https://doi.org/10.1175/bams-d-19-0022.1, 2020.

Liu, Z., Zhu, B., Zhu, C., Ruan, T., Li, J., Chen, H., Li, Q., Wang, X., Wang, L., Mu, Y., Collett, J., George, C., Wang, Y., Wang, X., Su, J., Yu, S., Mellouki, A., Chen, J., and Jiang, G.: Abundant nitrogenous secondary organic aerosol formation
285 accelerated by cloud processing, iScience, 26, 108317, https://doi.org/10.1016/j.isci.2023.108317, 2023.

Mandariya, A. K., Gupta, T., and Tripathi, S. N.: Effect of aqueous-phase processing on the formation and evolution of organic aerosol (OA) under different stages of fog life cycles, Atmos. Environ., 206, 60-71, https://doi.org/10.1016/j.atmosenv.2019.02.047, 2019.

Pailler, L., Deguillaume, L., Lavanant, H., Schmitz, I., Hubert, M., Nicol, E., Ribeiro, M., Pichon, J. M., Vaïtilingom, M., Dominutti, P., Burnet, F., Tulet, P., Leriche, M., and Bianco, A.: Molecular composition of clouds: a comparison between samples collected at tropical (Réunion Island, France) and mid-north (Puy de Dôme, France) latitudes, Atmos. Chem. Phys., 24, 5567-5584, https://doi.org/10.5194/acp-24-5567-2024, 2024.

Pasquier, J. T., David, R. O., Freitas, G., Gierens, R., Gramlich, Y., Haslett, S., Li, G., Schäfer, B., Siegel, K., Wieder, J., Adachi, K., Belosi, F., Carlsen, T., Decesari, S., Ebell, K., Gilardoni, S., Gysel-Beer, M., Henneberger, J., Inoue, J., Kanji, Z. A., Koike, M., Kondo, Y., Krejci, R., Lohmann, U., Maturilli, M., Mazzolla, M., Modini, R., Mohr, C., Motos, G., Nenes, A., Nicosia, A., Ohata, S., Paglione, M., Park, S., Pileci, R. E., Ramelli, F., Rinaldi, M., Ritter, C., Sato, K., Storelvmo, T., Tobo, Y., Traversi, R., Viola, A., and Zieger, P.: The Ny-Ålesund Aerosol Cloud Experiment (NASCENT): Overview and First Results, Bull. Amer. Meteorol. Soc., 103, E2533-E2558, https://doi.org/10.1175/bams-d-21-0034.1, 2022.

Qi, L., Chen, M., Stefenelli, G., Pospisilova, V., Tong, Y., Bertrand, A., Hueglin, C., Ge, X., Baltensperger, U., Prévôt, A. S. H., and Slowik, J. G.: Organic aerosol source apportionment in Zurich using an extractive electrospray ionization time-of-flight mass spectrometer (EESI-TOF-MS) – Part 2: Biomass burning influences in winter, Atmos. Chem. Phys., 19, 8037-8062, https://doi.org/10.5194/acp-19-8037-2019, 2019.

Sehested, K., Christensen, H. C., Hart, E. J., and Corfitzen, H.: Rates Of Reaction Of O$^-$, OH, And H With Methylated Benzenes In Aqueous Solution. Optical Spectra Of Radicals, Journal of Physical Chemistry, 79, 310-315, https://doi.org/10.1021/j100571a005, 1975.

Sun, W., Fu, Y., Zhang, G., Yang, Y., Jiang, F., Lian, X., Jiang, B., Liao, Y., Bi, X., Chen, D., Chen, J., Wang, X., Ou, J., Peng, P. a., and Sheng, G.: Measurement report: Molecular characteristics of cloud water in southern China and insights into aqueous-phase processes from Fourier transform ion cyclotron resonance mass spectrometry, Atmos. Chem. Phys., 21, 16631-16644, https://doi.org/10.5194/acp-21-16631-2021, 2021.

Sun, W., Zhang, G., Guo, Z., Fu, Y., Peng, X., Yang, Y., Hu, X., Lin, J., Jiang, F., Jiang, B., Liao, Y., Chen, D., Chen, J., Ou, J., Wang, X., Peng, P. a., and Bi, X.: Formation of In-Cloud Aqueous-Phase Secondary Organic Matter and Related Characteristic Molecules, J. Geophys. Res.-Atmos., 129, https://doi.org/10.1029/2023jd040355, 2024.

Tan, Y., Perri, M. J., Seitzinger, S. P., and Turpin, B. J.: Effects of Precursor Concentration and Acidic Sulfate in Aqueous Glyoxal-OH Radical Oxidation and Implications for Secondary Organic Aerosol, Environ. Sci. Technol., 43, 8105-8112, https://doi.org/10.1021/es901742f, 2009.

Tong, Y., Pospisilova, V., Qi, L., Duan, J., Gu, Y., Kumar, V., Rai, P., Stefenelli, G., Wang, L., Wang, Y., Zhong, H., Baltensperger, U., Cao, J., Huang, R.-J., Prévôt, A. S. H., and Slowik, J. G.: Quantification of solid fuel combustion and aqueous chemistry contributions to secondary organic aerosol during wintertime haze events in Beijing, Atmos. Chem. Phys., 21, 9859-9886, https://doi.org/10.5194/acp-21-9859-2021, 2021.

Zhang, X., Chen, Z. M., and Zhao, Y.: Laboratory simulation for the aqueous OH-oxidation of methyl vinyl ketone and methacrolein: significance to the in-cloud SOA production, Atmos. Chem. Phys., 10, 9551-9561, https://doi.org/10.5194/acp-10-9551-2010, 2010.

Zhao, Y., Hallar, A. G., and Mazzoleni, L. R.: Atmospheric organic matter in clouds: exact masses and molecular formula identification using ultrahigh-resolution FT-ICR mass spectrometry, Atmos. Chem. Phys., 13, 12343-12362, https://doi.org/10.5194/acp-13-12343-2013, 2013.